# STABLETOKEN: A NOISE-ROBUST SEMANTIC SPEECH TOKENIZER FOR RESILIENT SPEECHLLMS

**Yuhan Song**[1*] **Linhao Zhang**[2*†] **Chuhan Wu**[2] **Aiwei Liu**[2] **Wei Jia**[2]
**Houfeng Wang**[1†] **Xiao Zhou**[2]

[1] State Key Laboratory of Multimedia Information Processing,
  School of Computer Science, Peking University
[2] Basic Model Technology Center, WeChat AI, Tencent Inc.

✉ {songyuhan,wanghf}@pku.edu.cn zhanglinhao90@gmail.com

## ABSTRACT

Prevalent semantic speech tokenizers, designed to capture linguistic content, are surprisingly fragile. We find they are not robust to meaning-irrelevant acoustic perturbations; even at high Signal-to-Noise Ratios (SNRs) where speech is perfectly intelligible, their output token sequences can change drastically, increasing the learning burden for downstream LLMs. This instability stems from two flaws: a brittle single-path quantization architecture and a distant training signal indifferent to intermediate token stability. To address this, we introduce StableToken, a tokenizer that achieves stability through a consensus-driven mechanism. Its multi-branch architecture processes audio in parallel, and these representations are merged via a powerful bit-wise voting mechanism to form a single, stable token sequence. StableToken sets a new state-of-the-art in token stability, drastically reducing Unit Edit Distance (UED) under diverse noise conditions. This foundational stability translates directly to downstream benefits, significantly improving the robustness of SpeechLLMs on a variety of tasks. Our code and model are publicly available at https://github.com/Tencent/StableToken.

## 1 INTRODUCTION

The application of Large Language Models (LLMs) to the speech domain has given rise to a new class of powerful models: Speech Large Language Models (SpeechLLMs) (Hurst et al., 2024; Défossez et al., 2024; Zeng et al., 2024). These models rely on a discrete speech tokenizer to convert continuous audio into tokens sequences that the LLM can process. Among available methods, semantic tokenizers have been widely adopted, as their low-bitrate, semantically-aligned outputs are highly compatible with LLM architectures (Défossez et al., 2024; Zeng et al., 2024; Ding et al., 2025; Wu et al., 2025).

The design of semantic speech tokenizers has evolved from early self-supervised learning (SSL) methods (Hsu et al., 2021; Baevski et al., 2020) towards a more direct, supervised paradigm (Du et al., 2024a;b; Zeng et al., 2024). This modern paradigm centers on optimizing a VQ-based quantizer (Van Den Oord et al., 2017) with a direct, end-to-end objective such as automatic speech recognition (ASR) . This powerful combination has proven highly effective at producing semantically-rich and compact discrete representations, leading to the widespread adoption of supervised semantic tokenizers as the backbone of many modern SpeechLLMs (Zeng et al., 2024; Ding et al., 2025; Wu et al., 2025; Huang et al., 2025; Fang et al., 2025).

Despite their widespread adoption and apparent success, we find that these semantic tokenizers harbor a critical vulnerability: a profound lack of robustness. Contrary to their core design principle of encoding semantics, even imperceptible acoustic noise can induce drastic shifts in their discrete outputs (Figure 1). The general issue of tokenizer instability is also supported by recent findings on earlier non-VQ-based SSL tokenizers (Messica & Adi, 2024). This instability creates a damaging downstream effect: small acoustic changes trigger large token jumps, which break the crucial

---

[*]Equal contribution. Work conducted during Yuhan's internship at WeChat.
[†]Corresponding authors.

Figure 1: Illustration of StableToken: unlike traditional methods, StableToken yields consistent token sequences under minute perturbations with different Signal-to-Noise Ratios (SNRs). Robustness is measured by Unit Edit Distance (UED, ↓) between token sequences of the original and noise-perturbed audio. StableToken achieves significantly lower UED, indicating enhanced token stability.

speech–text alignment and pose a significant modeling challenge for the LLM, forcing it to learn from an inconsistent or even chaotic input stream. This inherent fragility is severely amplified by environmental noise, serving as a key cause for the performance degradation of SpeechLLMs in real-world conditions (Ma et al., 2025b; Zhang et al., 2025; Yang et al., 2024c; Jiang et al., 2025). We argue that enhancing tokenizer robustness is therefore a direct and promising path toward building more resilient models.

We pinpoint the fragility of semantic tokenizers to two fundamental weaknesses. First, an **architectural flaw**: these tokenizers rely on single-path quantization, a design that lacks fault tolerance. A minor perturbation near a quantization boundary is inevitably magnified into a completely different output token. This architectural vulnerability is then compounded by a **distant supervisory signal**: the standard ASR loss is indifferent to this intermediate token instability, as it only supervises the final transcribed text. This allows models to converge on solutions that are functionally correct but representationally fragile. The dual challenge of a brittle architecture and a distant supervisory signal necessitates a new tokenization paradigm.

Addressing this dual challenge is non-trivial. For the brittle architecture, an offline ensemble of models seems intuitive. However, this approach is untenable: (1) it prohibitively increases inference cost; (2) aggregating independently trained models is non-trivial, as their quantization boundaries are arbitrarily aligned; and (3) a token-level majority vote is too coarse. To tackle the distant supervisory signal, one might introduce a token-level consistency objective for clean and noisy input audios. Yet, this leads to unstable gradients when applied to discrete codes, making the model difficult to train. The failure of these straightforward approaches underscores the need for a more integrated paradigm.

We propose StableToken, which integrates a co-designed architecture and training strategy to overcome the dual challenges of architectural fragility and distant supervision. **Architecturally**, it introduces the voting-LFQ module—a multi-branch quantizer extended from the LFQ algorithm (Yu et al., 2023), with negligible inference overhead. Its core mechanism is a differentiable bit-level majority vote. During training, this enables a more fine-grained fusion of multi-branch information, leading to more stable and robust representation learning. At inference, this same mechanism provides profound error-correction, operating at the granular bit-level rather than the coarse token-level. This distinction is critical: not only does it ensure the final token remains correct when a minority of branches err due to noise, but it can even recover the token when a majority of branches fail at the token-level, as long as the underlying bit-level errors remain sparse.

This architectural robustness is further solidified by a tailored **training strategy**. We present the model with multiple "views" of an input—a clean version to a majority of branches and a perturbed version to a random minority—to create a stable reference. A consensus loss then leverages this reference to provide the explicit, intermediate supervision. The multi-branch architecture and multi-view training strategy are thus deeply intertwined: the architecture provides the necessary structure for the training signal, and the signal in turn unlocks the architecture's full potential.

We validate StableToken through comprehensive experiments. At the tokenizer level, it achieves a new state-of-the-art in **noise robustness**, slashing the Unit Edit Distance (UED) by over 60% relative (from 26.17% to 10.17%), all while maintaining top-tier **reconstruction fidelity**. This

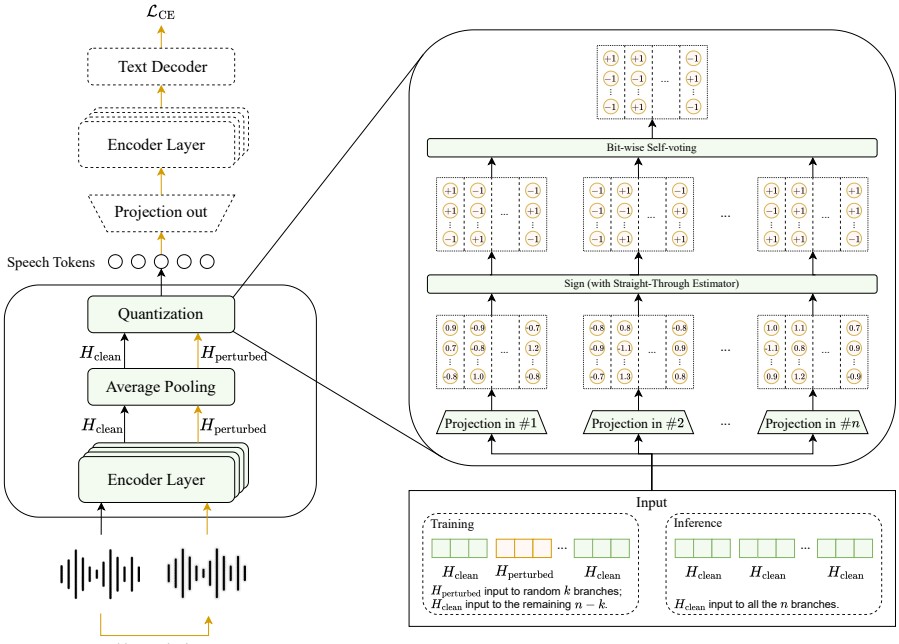

Figure 2: The architecture of StableToken. Our model replaces the standard single-path quantizer with a multi-branch **Voting-LFQ module**. The zoomed-in view shows $n$ parallel branches generating independent binary representations. A bit-wise majority vote then aggregates them into a single token. During our **Noise-Aware Consensus Training**, a randomly selected minority of branches receive perturbed inputs ($H_{\text{perturbed}}$), while the majority receive clean inputs ($H_{\text{clean}}$). A consensus loss forces the perturbed branches to align with the consensus. The yellow paths are only used during training.

foundational superiority translates directly to **downstream SpeechLLMs**. In speech understanding, the downstream models yield significant robustness gains that are especially pronounced under severe noise. The performance gap between StableToken and baselines widens dramatically as the noise level increases. Similarly, for speech generation, the enhanced token consistency simplifies the learning task, resulting in substantially superior synthesis quality of downstream models. These results confirm that improving tokenizer robustness is directly and highly effective for building more resilient speechLLMs.

## 2 METHODS

### 2.1 OVERALL STRUCTURE

Our approach, StableToken, is designed to overcome the fragility of prevailing VQ-based semantic tokenizers. These tokenizers often produce unstable token sequences in the presence of subtle noise, a vulnerability stemming from two core weaknesses: (1) a single-path architecture that lacks fault tolerance, and (2) a distant supervisory signal that fail to enforce representational invariance.

StableToken adopts the architectural paradigm, established in works like (Du et al., 2024a; Zeng et al., 2024; Du et al., 2024b) of embedding a semantic tokenizer within an end-to-end ASR model. However, our approach fundamentally enhances this design by introducing two synergistic innovations to address its inherent instabilities: (1) the Voting-LFQ Module, a multi-branch quantizer that builds in architectural robustness, and (2) Noise-Aware Consensus Training, a training strategy that explicitly enforces invariance to acoustic perturbations.

### 2.2 THE VOTING-LFQ MODULE

The foundation of StableToken is a novel quantizer architecture designed for intrinsic robustness. As shown in Figure 2, the pretrained speech encoder first processes the input speech into a sequence

of hidden states. These states are then downsampled via average pooling to produce a compact representation, $\mathbf{h} \in \mathbb{R}^D$, for each time step.

While traditional quantizers map $\mathbf{h}$ to a token in a single, brittle step, our **Voting Look-up-Free Quantizer (Voting-LFQ)** is founded on redundancy and consensus. It begins by creating $n$ independent "perspectives" of the input state $\mathbf{h}$ using $n$ parallel linear projection layers. Each branch $i \in \{1, \dots, n\}$ computes a projected vector $\mathbf{p}_i \in \mathbb{R}^d$:

$$\mathbf{p}_i = W_i \mathbf{h} + \mathbf{b}_i, \tag{1}$$

where $W_i \in \mathbb{R}^{d \times D}$ and $\mathbf{b}_i \in \mathbb{R}^d$ are the unique learnable parameters for that branch. Each projected vector is then binarized into $\mathbf{B}_i \in \{-1, +1\}^d$ using the non-differentiable sign function, i.e., $\mathbf{B}_i = \text{sign}(\mathbf{p}_i)$. We use the Straight-Through Estimator (STE) (Bengio et al., 2013) to enable end-to-end training.

During **training**, we aggregate these $n$ binary vectors in a bit-wise manner by averaging their values across branches for every dimension $j \in \{1, \dots, d\}$, resulting in a real-valued score:

$$(\mathbf{s}_{\text{final}})_j = \frac{1}{n} \sum_{i=1}^{n} (\mathbf{B}_i)_j. \tag{2}$$

Unlike the rigid assignment of a single bit ($+1$ or $-1$), these averaged scores can take nuanced values representing the confidence or consensus of all branches. This provides the model with richer feedback during optimization, helping it learn more robust and informative representations.

During **inference**, we perform one additional step to apply the sign function to these aggregated scores to obtain the final consensus-based binary vector:

$$(\mathbf{B}_{\text{final}})_j = \text{sign}\left((\mathbf{s}_{\text{final}})_j\right). \tag{3}$$

By using an odd number of parallel branches ($n$), we enforce a strict majority rule via a bit-wise vote, creating exceptional robustness against noise. This approach not only corrects errors when a minority of branches fail, but can also recover the true token even if a majority of branches are corrupted at the token-level. Recovery is possible as long as the underlying bit-level errors remain sparse. This resilience marks a significant advantage over fragile single-path quantizers and, in parallel, allows for more expressive representations during training.

Finally, by mapping its $-1$ and $+1$ entries to 0 and 1 respectively and treating the $\{0, 1\}$ representation as a binary number, the stabilized binary vector $\mathbf{B}_{\text{final}}$ is deterministically mapped to an integer index $k \in \{0, \dots, 2^d - 1\}$. This index serves as the final, robust speech token. It is worth noting that our Voting-LFQ structure introduces negligible additional parameters and computational overhead during inference. A detailed complexity analysis is provided in Appendix B.6.

## 2.3 Noise-Aware Consensus Training

The Voting-LFQ architecture enables our novel training paradigm, designed to explicitly instill representational invariance. Our goal is to make the tokenizer robust to noise without degrading its performance on clean inputs.

The core mechanism works as follows: during each forward pass, for a given input audio $\mathbf{w}$, we generate a perturbed audio sample $\mathbf{w}' = \mathcal{A}(\mathbf{w})$, where $\mathcal{A}(\cdot)$ is a stochastic augmentation function applied at the waveform level (e.g., adding Gaussian noise; further details in Appendix B.3). Both $\mathbf{w}$ and $\mathbf{w}'$ are separately processed by the encoder to produce two corresponding hidden states, $\mathbf{h}$ and $\mathbf{h}'$. we then randomly select a minority subset of $k$ branches (where $k < n/2$) to receive the perturbed hidden state $\mathbf{h}'$, while the remaining $n - k$ majority branches receive the clean hidden state $\mathbf{h}$.

This setup allows the model to perform self-stabilization. To enforce this, we introduce the **consensus loss** ($\mathcal{L}_{\text{consensus}}$) which encourages all branches, whether they see a clean or noisy input, to produce similar pre-quantization representations. We compute a dynamic, "online" target $\bar{\mathbf{p}}_{\text{all}}$ by averaging the pre-quantization vectors $\mathbf{p}_i$ from **all** $n$ branches. The loss then penalizes the deviation of each branch from this global average:

$$\mathcal{L}_{\text{consensus}} = \frac{1}{n} \sum_{i=1}^{n} \|\mathbf{p}_i - \bar{\mathbf{p}}_{\text{all}}\|_2^2, \quad \text{where} \quad \bar{\mathbf{p}}_{\text{all}} = \frac{1}{n} \sum_{j=1}^{n} \mathbf{p}_j. \tag{4}$$

By optimizing this objective, the clean-majority branches act as a stable anchor for the global average $\bar{\mathbf{p}}_{\text{all}}$, preventing it from being corrupted by the noisy inputs. Consequently, the noisy-minority branches are forced to learn representations that align with the clean consensus, effectively learning to ignore the perturbations. Optimizing on the continuous vectors $\mathbf{p}_i$ provides a smoother and more effective gradient signal than working with the binarized $\mathbf{b}_i$.

## 2.4 FINAL TRAINING OBJECTIVE

The complete training objective for StableToken combines the ASR task loss with our consensus loss and standard LFQ regularization terms. The primary task is optimized via a Cross-Entropy loss ($\mathcal{L}_{\text{ASR}}$) on the ground-truth transcripts. Following the LFQ framework (Yu et al., 2023), we also include a **commitment loss** ($\mathcal{L}_{\text{commitment}}$) to encourage the hidden states to stay close to the quantized representations, and a **codebook entropy loss** ($\mathcal{L}_{\text{codebook}}$) to promote uniform usage of the discrete codes. The final, composite loss function is a weighted sum:

$$\mathcal{L}_{\text{total}} = \mathcal{L}_{\text{ASR}} + \lambda_1 \mathcal{L}_{\text{consensus}} + \lambda_2 \mathcal{L}_{\text{commitment}} + \lambda_3 \mathcal{L}_{\text{codebook}}, \tag{5}$$

where $\lambda_1, \lambda_2$, and $\lambda_3$ are scalar hyperparameters that balance the influence of each component.

## 3 EXPERIMENTAL SETUP

Our experiments are structured to comprehensively validate StableToken from three perspectives. We first demonstrate its core superiority at the **tokenizer level**, establishing a new state-of-the-art in noise robustness without compromising reconstruction quality (§4.1). We then show that this fundamental stability translates directly into significant performance gains in **diverse downstream SpeechLLM tasks**, including ASR, Speech Emotion Recognition (SER), and Text-to-speech (TTS) (§4.2). Finally, we dissect the model through **ablation studies, qualitative analysis, and a case study** to verify the contribution of each design component and provide insight into its inner workings (§4.3).

**Tokenizer Training.** Our StableToken model is built upon an encoder-decoder architecture initialized from `whisper-large-v3` (Radford et al., 2023), with our Voting-LFQ module inserted into the encoder's mid-point. The tokenizer is pre-trained on a diverse 150k-hour speech corpus. Our tokenizer vocabulary size is set to 8192 (corresponding to $d = 13$), and the frame rate is 25Hz. For our main experiments, the number of voters is set to $N = 5$, a choice justified by our analysis in Section 4.3. Full details on training data hyperparameters are provided in Appendix B.

**Baseline Models.** We benchmark StableToken against a comprehensive suite of SOTA models across three categories: SSL-based, distilled, and supervised tokenizers. For all baselines, we use their officially released models. The detailed list of baseline models can be found in Appendix D.

**Tokenizer-Level Evaluation.** We assess the tokenizer's intrinsic properties. **Robustness** is measured by Unit Edit Distance (UED%, ↓) (Messica & Adi, 2024) on the FLEURS (Conneau et al., 2023) benchmark under various synthetic perturbations and real-world noise conditions, which include challenging out-of-domain (OOD) noise. A detailed description of the noise profiles is in Appendix E. **Fidelity** is measured by Word Error Rate (WER%, ↓) and Mean Opinion Score (MOS, ↑) on the LibriSpeech (Panayotov et al., 2015) and SEED (Anastassiou et al., 2024) benchmarks.

**Downstream Task Evaluation.** For downstream tasks, we follow a controlled, isogenic setup to ensure fair comparison. Each tokenizer is integrated into a SpeechLLM framework using a pre-trained `Qwen2.5-3B` (Yang et al., 2024a) backbone, which is then fine-tuned using a prompt-based paradigm (Zeng et al., 2025). We evaluate on three tasks: (1) **ASR:** Assessed on noise-augmented LibriSpeech (Panayotov et al., 2015) and the CHiME-4 (Vincent et al., 2017) benchmark using WER (%); (2) **SER:** Assessed on a noise-augmented version of the ESD (Zhou et al., 2022) test set using classification accuracy (%); (3) **TTS:** Assessed on the SEED-TTS (Anastassiou et al., 2024) benchmark using both WER (%) and MOS. The aggregated training datasets, fine-tuning hyperparameters, and prompts for each task are detailed in Appendix F.

Table 1: Noise robustness comparison across different semantic tokenizers. Results are reported in UED% (↓) under synthetic perturbation (Gaussian, Pink, Brown, Bit Crush) and real noise conditions. It is worth noting that a comparison is most meaningful between tokenizers of the same type. For a more comprehensive evaluation, we also include SSL and semantic distilled tokenizers as baselines.

| Model | #C | Frame Rate | Codebook Size | Gauss. Noise | Pink Noise | Brown Noise | Bit Crush | Real Noise | Real (OOD) | Avg. |
|---|---|---|---|---|---|---|---|---|---|---|
| **SSL Semantic Tokenizer** | | | | | | | | | | |
| HuBERT-500 (Hsu et al., 2021) | 1 | 50Hz | 500 | 26.42 | 20.38 | 18.82 | 18.02 | 18.48 | 19.18 | 20.22 |
| NAST (Messica & Adi, 2024) | 1 | 50Hz | 200 | 18.67 | 15.78 | 15.26 | 14.95 | 18.69 | 19.07 | 17.07 |
| R-Spin (Chang & Glass, 2024) | 1 | 50Hz | 2048 | 21.56 | 17.08 | 15.47 | 14.95 | 15.08 | 14.75 | 16.48 |
| **Semantic Distilled Tokenizer** | | | | | | | | | | |
| SpeechTokenizer (Zhang et al., 2023) | 1 | 50Hz | 1024 | 37.39 | 28.05 | 28.06 | 21.38 | 22.33 | 23.09 | 26.72 |
| | 3 | 50Hz | 1024 | 55.69 | 54.90 | 59.84 | 35.29 | 33.16 | 33.67 | 45.43 |
| | 8 | 50Hz | 1024 | 72.74 | 72.72 | 75.91 | 54.01 | 48.43 | 48.63 | 62.07 |
| X-Codec (Ye et al., 2025) | 1 | 50Hz | 1024 | 53.54 | 43.85 | 40.17 | 36.95 | 27.82 | 28.78 | 38.52 |
| | 3 | 50Hz | 1024 | 71.76 | 59.95 | 57.88 | 50.26 | 41.25 | 42.44 | 53.92 |
| | 8 | 50Hz | 1024 | 84.46 | 77.31 | 76.49 | 68.47 | 59.89 | 62.28 | 71.48 |
| Mimi (Défossez et al., 2024) | 8 | 12.5Hz | 2048 | 72.68 | 59.82 | 60.19 | 43.58 | 41.66 | 42.62 | 53.43 |
| **Supervised Semantic Tokenizer** | | | | | | | | | | |
| GLM-4-Voice-Token. (Zeng et al., 2025) | 1 | 12.5Hz | 16384 | 42.44 | 32.12 | 30.22 | 25.53 | 27.67 | 28.62 | 31.10 |
| $\mathcal{S}^3$ Tokenizer (Du et al., 2024a) | 1 | 25Hz | 4096 | 35.40 | 27.09 | 25.45 | 20.64 | 23.88 | 24.58 | 26.17 |
| | 1 | 50Hz | 4096 | 46.05 | 35.90 | 33.46 | 27.20 | 27.70 | 28.21 | 33.09 |
| CosyVoice2 (Du et al., 2024b) | 1 | 25Hz | 6561 | 54.67 | 42.57 | 39.96 | 30.87 | 31.76 | 32.13 | 38.66 |
| **StableToken (Ours)** | 1 | 25Hz | 8192 | **12.93** | **9.76** | **9.37** | **7.32** | **10.65** | **10.96** | **10.17** |

## 4 RESULTS

### 4.1 TOKENIZER-LEVEL PERFORMANCE

#### 4.1.1 SUPERIOR NOISE ROBUSTNESS

As shown in Table 1, StableToken establishes a new state-of-the-art in noise robustness. It achieves an average UED of **10.17%**, a dramatic improvement over both the best supervised baseline ($\mathcal{S}^3$ Tokenizer, 26.17%) and the top-performing robust SSL-based model (R-Spin, 16.48%). Crucially, this strong performance holds even on out-of-distribution (OOD) real-world noise not seen during training, demonstrating the excellent generalization of our method. Furthermore, this outperformance is achieved using a significantly larger vocabulary than conventional tokenizers. This makes the result even more significant, as a larger vocabulary creates a finer-grained decision space, making the task of maintaining token-level invariance inherently more challenging. This substantial performance gap underscores the effectiveness of our co-designed architecture and training strategy.

#### 4.1.2 EXCELLENT RECONSTRUCTION QUALITY

Table 2: Reconstruction results measured by WER (↓) and MOS (↑) on LibriSpeech (Panayotov et al., 2015) and SEED (Anastassiou et al., 2024) benchmarks.

| Model | #C | Frame Rate | BPS | WER ↓ | | | | MOS ↑ | | | |
|---|---|---|---|---|---|---|---|---|---|---|---|
| | | | | LS-clean | LS-other | SEED en | SEED zh | LS-clean | LS-other | SEED en | SEED zh |
| GLM-4-Voice-Token. (Zeng et al., 2025) | 1 | 12.5Hz | 175 | 4.04 | 9.33 | 3.54 | 3.23 | 4.07 | **3.99** | **4.16** | 4.10 |
| $\mathcal{S}^3$ Tokenizer (Du et al., 2024a) | 1 | 25Hz | 300 | 5.78 | 13.38 | 5.91 | 4.26 | 3.40 | 3.31 | 3.40 | 3.31 |
| CosyVoice2 (Du et al., 2024b) | 1 | 25Hz | 325 | 4.25 | 9.68 | 4.34 | 2.75 | 3.36 | 3.25 | 3.31 | 3.58 |
| **StableToken (Ours)** | 1 | 25Hz | 325 | **3.84** | **7.99** | **3.44** | **2.62** | **4.09** | 3.83 | 4.01 | **4.18** |

To evaluate reconstruction quality, we follow the methodology of Du et al. (2024a;b); Zeng et al. (2024) and train a flow matching model to synthesize audio from our speech tokens. The results, shown in Table 2, demonstrate that the leap in noise robustness does not compromise the tokenizer's fundamental quality. StableToken delivers state-of-the-art reconstruction performance, evidenced by its exceptional Word Error Rate (WER) and Mean Opinion Scores (MOS). These results validate

StableToken as a versatile tokenizer that excels in both resilience and fidelity. Details on the audio reconstruction setup are provided in Appendix B.5.

## 4.2 Downstream SpeechLLM Performance

The ultimate measure of a tokenizer's utility is its impact on downstream tasks. We find that StableToken's intrinsic robustness consistently translates to superior performance in ASR, SER, and TTS, especially in challenging, noisy conditions.

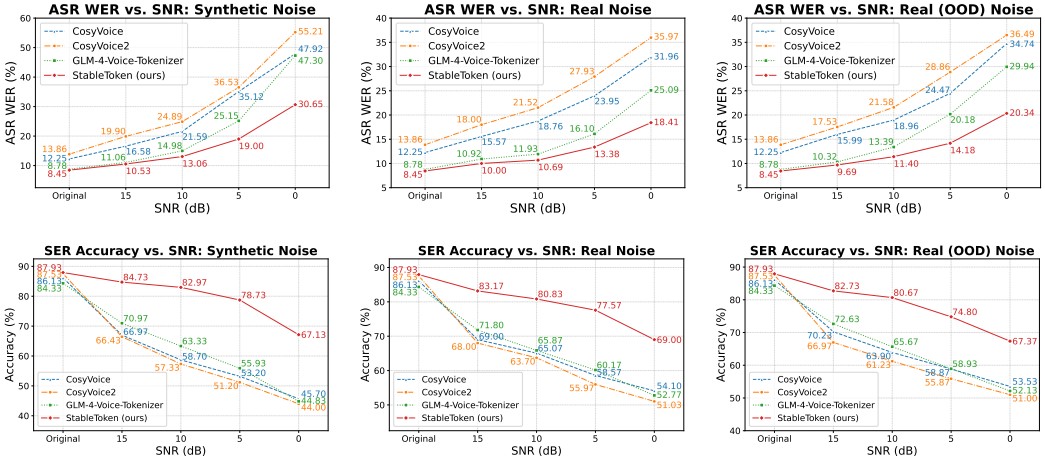

Figure 3: Performance of downstream SpeechLLMs under various noise conditions and SNR levels. **(Top Row)** ASR performance, measured in Word Error Rate (WER, ↓). **(Bottom Row)** SER performance, measured in Accuracy (↑). In both tasks, StableToken consistently demonstrates superior robustness, with the performance gap widening as noise severity increases.

### 4.2.1 Robust ASR Performance

StableToken significantly contributes to a robust downstream ASR model. Figure 3 shows that while all systems perform comparably on clean audio, the performance gap widens dramatically as noise increases. Under the most severe OOD real-world noise at 0dB SNR, the model with StableToken achieves a WER of 20.34%, a relative reduction of over 30% compared to the baseline's 29.94%.

This robustness generalizes to complex acoustic scenes. On the CHiME-4 (Vincent et al., 2017) benchmark (Table 3), the StableToken-based system achieves WERs of 35.90% (test-real set) and 30.61% (test-simulated set), marking relative reductions of approximately 30% over the next-best baseline. This confirms that token-level stability is a direct driver of downstream model resilience.

### 4.2.2 Robust SER Performance

Figure 3 shows that the StableToken-based model consistently achieves higher classification accuracy across all noise types and levels. While performance is similar across all tokenizers on clean audio, the StableToken-based model's accuracy degrades much more slowly as noise increases, demonstrating greater robustness in isolating emotional cues from corrupted audio.

### 4.2.3 Superior TTS Performance

As shown in Table 3, StableToken delivers superior TTS performance, with significantly lower WER on both subsets, as well as an improved MOS on SEED-TTS$_{ZH}$ and competitive performance on SEED-TTS$_{EN}$. The reduction in WER confirms that our tokenizer enables more intelligible speech synthesis that faithfully reproduces the intended text. Concurrently, the MOS score demonstrates high naturalness and auditory quality of the synthetic speech. Together, these results strongly support StableToken as an effective and information-rich representation for speech synthesis.

Table 3: Downstream SpeechLLMs performance comparison on ASR (CHiME-4) and TTS (SEED-TTS) benchmarks. In both tasks, integrating StableToken into the downstream SpeechLLM leads to substantially improved performance, demonstrating its noise robustness and versatility.

| Tokenizer | LLM-base | ASR | | | | TTS | | | |
| | | Dev Set | | Test Set | | SEED-TTS$_{EN}$ | | SEED-TTS$_{ZH}$ | |
| | | Real | Sim. | Real | Sim. | WER↓ | MOS↑ | WER↓ | MOS↑ |
|---|---|---|---|---|---|---|---|---|---|
| CosyVoice | Qwen2.5-3B | 38.66 | 40.82 | 54.63 | 47.71 | 7.80 | 3.52 | 8.73 | 3.47 |
| CosyVoice2 | Qwen2.5-3B | 43.91 | 48.39 | 59.83 | 55.01 | 7.22 | 3.75 | 9.89 | 3.37 |
| GLM-4-Voice | Qwen2.5-3B | 36.92 | 36.38 | 51.08 | 43.09 | 6.19 | **4.19** | 5.26 | 3.85 |
| StableToken | Qwen2.5-3B | **25.56** | **25.36** | **35.90** | **30.61** | **4.43** | 4.12 | **3.02** | **4.08** |

In summary, across understanding (ASR, SER) and generation (TTS), StableToken consistently enables stronger, more reliable downstream performance. This dual advantage affirms its effectiveness as a powerful, versatile foundation for real-world speech systems.

## 4.3 ANALYSIS

### 4.3.1 COMPONENT ABLATION STUDY

Table 4: Sequential ablation study of StableToken. We jointly evaluate tokenizer robustness (UED ↓) and semantic preservation (ASR WER ↓). Results show that each component contributes to robustness, with the full model providing optimal stability and semantic fidelity. ASR is measured on the validation set during tokenizer training.

| Model Configuration | Tokenizer Robustness (UED% ↓) | | | | | | ASR (WER% ↓) | |
| | Gauss. Noise | Brown Noise | Pink Noise | Bit Crush | Real Noise | Real (OOD) | LS-Clean | LS-Other |
|---|---|---|---|---|---|---|---|---|
| **StableToken (Full)** | **12.93** | **9.76** | **9.37** | **7.32** | **10.65** | **10.96** | **2.03** | **4.68** |
| ↱ *w/o Consensus Loss* | 24.80 | 19.06 | 17.81 | 14.03 | 16.97 | 17.43 | **2.03** | 4.88 |
| ↱ *w/o Noise-Aware Training* | 30.77 | 23.05 | 21.30 | 17.32 | 20.95 | 21.51 | 2.19 | 5.52 |
| ↱ *w/o Multi-Branch* | 34.53 | 25.44 | 24.58 | 19.83 | 23.68 | 24.47 | 2.39 | 5.85 |

Our sequential ablation study, presented in Table 4, confirms that each component of StableToken is critical for its performance. First, removing the *Consensus Loss* causes the significant degradation in token robustness (e.g., UED on Real OOD noise increases from 10.96% to 17.43%), which underscores the importance of enforcing explicit agreement between branches. Subsequently, removing the *Noise-Aware Training* further harms performance, particularly the preservation of semantic content (WER on LibriSpeech-Other increases from 4.88% to 5.52%). Finally, reverting to a single-branch baseline results in the poorest performance overall. This highlights the *Multi-Branch architecture*'s dual role: it is the structural enabler for our training strategy and acts as an effective ensemble at inference, mitigating quantization errors common in single-path designs (Ma et al., 2025a).

### 4.3.2 ANALYSIS OF VOTER COUNT ($N$)

To determine the optimal number of voters that best balances performance and computational cost, we conduct preliminary training runs for each configuration. The results in Table 5 show a clear trend: increasing $N$ from 3 to 5 yields substantial improvements in both robustness and semantic preservation. However, a further increase to $N = 7$ offers only marginal gains that do not justify the added computational overhead. We therefore select $N = 5$ as the optimal configuration for all experiments. Analysis of parameters and FLOPs for different $N$ can be found in Appendix B.6.

Table 5: Impact of the number of voters ($N$) on tokenizer robustness and semantic preservation.

| | Tokenizer Robustness (UED% ↓) | | | | | | ASR (WER% ↓) | |
|---|---|---|---|---|---|---|---|---|
| **Number of Voters** ($N$) | **Gauss. Noise** | **Brown Noise** | **Pink Noise** | **Bit Crush** | **Real Noise** | **Real (OOD)** | **LS-Clean** | **LS-Other** |
| $N = 3$ | 20.66 | 15.42 | 14.44 | 11.55 | 14.89 | 15.27 | 2.24 | 5.47 |
| $N = 5$ | 18.68 | 13.87 | 13.11 | 10.50 | 14.06 | 14.49 | 2.22 | 5.38 |
| $N = 7$ | 18.10 | 13.35 | 12.51 | 9.84 | 13.79 | 14.11 | 2.36 | 5.52 |

Table 6: Case study on error correction via bit-wise voting.

| Output Source | Token @ Pos. 68 *(Vote on Bit #4)* | Token @ Pos. 80 *(Vote on Bit #5, #7)* | Token @ Pos. 105 *(Vote on Bit #3)* | Token @ Pos. 114 *(Vote on Bit #2, #6)* |
|---|---|---|---|---|
| Clean Reference | 5517 ...10001101 | 3485 ...10011101 | 2920 ...01101000 | 6939 ...00011011 |
| Voter 1 (Noisy) | 5533 ...10011101 | 3485 ...10011101 | 2920 ...01101000 | 6939 ...00011011 |
| Voter 2 (Noisy) | 5517 ...10001101 | 3517 ...10111101 | 2912 ...01100000 | 6943 ...00011111 |
| Voter 3 (Noisy) | 5517 ...10001101 | 3517 ...10111101 | 2920 ...01101000 | 6939 ...00011011 |
| Voter 4 (Noisy) | 5517 ...10001101 | 3485 ...10011101 | 2920 ...01101000 | 7003 ...01011011 |
| Voter 5 (Noisy) | 5533 ...10011101 | 3357 ...00011101 | 2920 ...01101000 | 6939 ...00011011 |
| Final Voted Output | 5517 Bit #4: 3 vs 2 → 0 | 3485 Bit #5: 3 vs 2 → 0 Bit #7: 4 vs 1 → 1 | 2920 4 vs 1 → 1 | 6939 Bit #2: 4 vs 1 → 0 Bit #6: 4 vs 1 → 0 |

### 4.3.3 CASE STUDY

Table 6 provides a case study illustrating the error correction capability of the voting-LFQ module. For instance, at position 80, noise causes three voters to generate erroneous tokens. Specifically, Voters 2 and 3 flip bit #5, while Voter 5 flips bit #7. Despite **most voters** predicting incorrect tokens, the voting mechanism operating at the bit level allows for correct recovery. For bit #5, the correct value '0' wins by a 3-to-2 majority, and for bit #7, the correct value '1' wins by a 4-to-1 majority, successfully reconstructing the original token (3485). Similar corrections occur at positions 68, 105 and 114. This case study highlights a key advantage of StableToken: its resilience does not depend on every branch being perfect, but on the collective ability to override sparse bit-flip errors.

## 5 RELATED WORK

**Semantic Speech Tokenizers**   The evolution of LLMs has driven the transition of spoken dialogue models from traditional pipelines to end-to-end SpeechLLMs (Zhang & Wang, 2019; Zhang et al., 2020; Jacqmin et al., 2022; Lee et al., 2021; Fang et al., 2024; Défossez et al., 2024; Wang et al., 2024), with semantic tokenizers becoming increasingly crucial. The design of semantic tokenizers has evolved through several distinct paradigms. Early approaches utilized self-supervised learning (SSL) to derive discrete units from unlabeled data (Hsu et al., 2021; Baevski et al., 2020; Chen et al., 2022; Chung et al., 2021; Conneau et al., 2021; Chiu et al., 2022; Baevski et al., 2019; Liu et al., 2023; Gat et al., 2023; Huang et al., 2022; Lodagala et al., 2023; Chang et al., 2023). The vast majority of tokens produced by these methods are designed for discriminative tasks. It is reported that discretized SSL tokens primarily encode phonetic information, causing high Gross Pitch Error (GPE) when paired with a vocoder for audio generation, making them unsuitable for end-to-end SpeechLLMs (Sicherman & Adi, 2023; Polyak et al., 2021; Mousavi et al., 2024; Guo et al., 2025).

A second category employs a hybrid approach, enhancing an acoustic tokenizer with semantic distillation to balance acoustic fidelity and semantic content (Zhang et al., 2023; Ye et al., 2025; Défossez et al., 2024; Siahkoohi et al., 2022; Yang et al., 2024b). This design enables strong performance on both generative and discriminative tasks. However, their integration with downstream large language models (LLMs) is hampered by several significant challenges. First, to preserve high fidelity, these methods tend to encode excessive acoustic details, which results in a high bits-per-second (BPS) rate. This high data rate generates longer token sequences, thereby increasing the computational load and impairing training efficiency. Furthermore, their reliance on Residual Vector Quantization (RVQ) produces hierarchical tokens that are inherently incompatible with the flat input structure expected by most LLMs. Collectively, the high data rate, the structural mismatch, and the overhead of processing superfluous acoustic information present substantial obstacles to their application in modern SpeechLLMs.

More recently, a third and more direct paradigm has gained traction: fully supervised training. Given that the primary goal is to capture semantic and phonetic information, this approach directly uses an Automatic Speech Recognition (ASR) objective for supervision. The process involves quantizing the intermediate representations of a powerful ASR encoder and optimizing the model with an ASR loss, ensuring the resulting discrete tokens directly represent linguistic units (Zeng et al., 2025; Du et al., 2024a;b). Subsequently, a downstream vocoder is trained to convert these discrete tokens into mel-spectrograms for speech synthesis. This tokenizer design is foundational to the current state-of-the-art end-to-end SpeechLLMs, underscoring its effectiveness and growing adoption. Interestingly, research has revealed that while the ASR objective targets linguistic content, the resulting tokens retain sufficient extra-phonetic information, such as prosody. This is likely because the ASR encoder implicitly learns to model prosodic features as they serve as valuable auxiliary cues for achieving high transcription accuracy. This retained information allows an integrated LLM to generate highly expressive synthesis and convey complex emotions. Consequently, this design's ability to support expressive generation has made it a foundational choice for state-of-the-art SpeechLLMs (Zeng et al., 2024; Ding et al., 2025).

**Noise Robustness**   Ensuring the stability of discrete speech tokens in the presence of noise is critical for the performance of modern Speech Language Models (SLMs). However, this issue has been largely overlooked compared to the extensive research focused on improving the robustness of the Automatic Speech Recognition (ASR) model itself (Wang et al., 2022; Tjandra et al., 2023; Eickhoff et al., 2023; Gong et al., 2023; Ahn et al., 2025). Recently, two studies have begun to address this gap by investigating the noise robustness of traditional SSL-based speech tokenizers.

R-SPIN (Chang & Glass, 2024) addresses this by learning speaker- and noise-invariant discrete units through a data-efficient self-supervised framework. It extends the speaker-invariant clustering of Spin by using an additional noise-perturbed view of the input and an auxiliary loss that predicts "acoustic pieces," which are phoneme-aligned pseudo-labels, to prevent model collapse and ensure the resulting discrete units represent pure linguistic content . In contrast, NAST (Messica & Adi, 2024) proposes an architecture designed explicitly for robust tokenization, consisting of a predictor, a residual encoder, and a decoder. Its training is governed by a combination of a reconstruction loss, a diversity loss to encourage codebook usage, and a crucial robustness loss that penalizes changes in the predicted token distribution between clean and noise-augmented versions of the same utterance, thereby directly optimizing for token-level stability. Liu et al. (2025) introduce slice-consistency and perturbation-consistency constraints to mitigate discrete representation inconsistency, but their approach targets acoustic tokenizers (rather than semantic tokenizers), which prioritize audio detail reconstruction. Therefore, noise invariance is less meaningful in their context, making their work fundamentally different from ours.

## 6   CONCLUSION

We introduce StableToken, a novel tokenizer designed to solve the critical instability of existing semantic tokenizers in noisy environments. By employing a multi-branch architecture and a consensus mechanism with bitwise voting, StableToken achieves state-of-the-art token stability. This stability directly translates to significant improvements in the robustness of downstream SpeechLLMs.

## ACKNOWLEDGMENTS

This work was supported by National Natural Science Foundation of China (62576010). The corresponding authors are Linhao Zhang and Houfeng Wang.

## REPRODUCIBILITY STATEMENT

To facilitate reproducibility of our work, we have provided detailed descriptions of the datasets, hyperparameters, and other experimental details used in our study in Section 3 and Appendix B, E, F. Our code and model checkpoint will be released publicly upon acceptance to further support reproducibility and foster future research.

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

## A LARGE LANGUAGE MODEL (LLM) USAGE STATEMENT

In accordance with the conference policies on Large Language Model (LLM) usage, we hereby disclose the following: After completing the initial draft of this paper, we utilized an LLM to enhance grammar and polish the writing of this manuscript. No new research ideas, experimental designs, or scientific content were generated by the LLM. All scientific contributions, analyses, and conclusions presented in this work are solely those of the authors. We take full responsibility for the content of this paper, including all sections that have been revised or improved with LLM assistance. The LLM is not an author and did not contribute to the research ideation or substantive scientific writing.

This statement is provided to ensure transparency and compliance with the conference's policies on LLM usage.

## B DETAILS OF STABLETOKEN

### B.1 TRAINING DATASETS FOR STABLETOKEN

We train our StableToken model on hundreds of thousands of hours of both open-source data and in-house data. All open-source datasets used in this work are listed in Table 7.

Table 7: Summary of datasets used for training StableToken

| Dataset | Duration (#hours) | Task | Language(s) |
|---|---|---|---|
| LibriSpeech (Panayotov et al., 2015) | 960 | ASR | English |
| Multilingual LibriSpeech (Pratap et al., 2020) | 27,322 | ASR | English |
| The People's Speech (Galvez et al., 2021) | 5,568 | ASR | English |
| GigaSpeech (Chen et al., 2021) | 10,000 | ASR | English |
| Yodas (Li et al., 2023) | 29,155 | ASR | English |
| Hi-Fi TTS (Bakhturina et al., 2021) | 292 | ASR | English |
| VCTK (Veaux et al., 2017) | 44 | ASR | English |
| LibriTTS (Zen et al., 2019) | 586 | ASR | English |
| VoiceAssistant-400K (Xie & Wu, 2024) | 679 | ASR | English |
| AISHELL-1 (Bu et al., 2017) | 150 | ASR | Chinese |
| WenetSpeech (Zhang et al., 2022) | 10,005 | ASR | Chinese |
| Common Voice (Ardila et al., 2019) | 2,133 | ASR | English, Chinese |
| Emilia (He et al., 2024) | 96,750 | ASR | English, Chinese |

### B.2 TRAINING HYPERPARAMETERS FOR STABLETOKEN

Table 8 summarizes the main hyperparameters used throughout StableToken training.

Table 8: Hyperparameters used for training StableToken

| Hyperparameter | Value |
|---|---|
| optimizer_type | AdamW |
| lr_scheduler | OneCycleLR |
| max_lr | 1.5e-5 |
| warmup_steps | 1000 |
| weight_decay | 0.01 |
| grad_clip | 1.0 |
| consensus_loss_weight ($\lambda_1$) | 0.25 |
| commitment_loss_weight ($\lambda_2$) | 0.25 |
| codebook_entropy_loss_weight ($\lambda_3$) | 1.0 |

### B.3 Details of Stochastic Perturbations During Training

We detail the construction and parameterization of the stochastic augmentation function $\mathcal{A}(\cdot)$ as introduced in Section 2.3. For each sample, one perturbation type is randomly selected from the following five categories: Gaussian Noise, Pink Noise, Brown Noise, Bit Crush Distortion, and Real-world Noise. The intensity of the selected perturbation is then uniformly sampled from a predefined range specific to each type, as summarized in Table 9. For real-world noise, an additional random selection of a noise audio clip is performed. The pool of noise clips consists of samples from the AudioCaps (Kim et al., 2019), FSD50k (Fonseca et al., 2021), and ESC-50 (Piczak, 2015) datasets. Notably, the ESC-10 subset of ESC-50 is excluded from training and reserved exclusively for evaluation as out-of-domain real-world noise.

Table 9: Perturbation types and their corresponding intensity ranges utilized during training.

| Perturbation Type | Intensity Range |
|---|---|
| Gaussian Noise | $16 \leq \text{SNR} \leq 30$ |
| Pink Noise | $16 \leq \text{SNR} \leq 24$ |
| Brown Noise | $12 \leq \text{SNR} \leq 24$ |
| Bit Crush Distortion | $8 \leq \text{Bit Depth} \leq 14$ |
| Real-world Noise | $12 \leq \text{SNR} \leq 24$ |

### B.4 Discussion of Consensus Objective Loss Choices

We choose the $L_2$ loss (Mean Squared Error Loss) for the consensus objective in Eq. 4 due to several considerations:

First, $L_2$ loss offers a simple and direct way to minimize the differences among multiple branches. At the same time, its form is also naturally compatible with the existing commitment loss in quantization-based models, thus facilitating stable optimization and consistent gradient behavior across objectives. Furthermore, by averaging representations from all branches, the resulting target inherently incorporates a form of confidence weighting, as outlier branch results are diluted in the mean.

Second, since all branches originate from the same underlying input, with only noise perturbations in minority branches, the goal is not to increase inter-class separation as in contrastive learning, but to enforce similarity across the noisy and clean versions. The L2 loss directly minimizes the Euclidean distance between the branches and their consensus, effectively encouraging robust invariance without introducing additional factors or requiring extra sampling. Moreover, cosine similarity is less sensitive to the number of bit flips in high-dimensional binary codes, especially when representations are already close, such as the flip of a single bit, which corresponds to small changes in angle and often results in diminished gradient signals and less effective correction of localized errors.

### B.5 Audio Reconstruction Details

Table 10: Summary of datasets used for training the flow matching model

| Dataset | Language(s) |
|---|---|
| Librilight (Kahn et al., 2020) | English |
| WenetSpeech (Zhang et al., 2022) | Chinese |
| Yodas2 (Li et al., 2023) | English, Chinese |
| Emilia (He et al., 2024) | English, Chinese |

Following the framework of CosyVoice (Du et al., 2024a) and GLM-4-Voice (Zeng et al., 2025), we train a flow matching model to reconstruct audio from speech tokens. The model takes as input the speech token representations and produces Mel spectrograms. Finally, a HiFi-GAN vocoder (Kong et al., 2020) converts the generated Mel spectrograms into the speech waveforms. The datasets for training the flow matching model is listed in Table 10 (excluding our in-house datasets, which comprise both English and Chinese speech data), and the training hyperparameters in Table 11.

Table 11: Hyperparameters used for training the flow matching model

| Hyperparameter | Value |
|---|---|
| optimizer_type | Adam |
| lr_scheduler | WarmupLR |
| learning_rate | 3.0e-4 |
| warmup_steps | 25000 |
| grad_clip | 1.5 |
| batch_type | dynamic |
| max_frames_in_batch | 10000 |

## B.6 ANALYSIS ON COMPUTATIONAL EFFICIENCY

### B.6.1 COMPUTATIONAL OVERHEAD OF DIFFERENT VOTER COUNTS

To further explore the computational overhead brought by increasing the number of voters ($N$), we present the model parameter counts and floating-point operations (FLOPs) for models with $N = 1, 3,$ 5, and 7 in Table 12. Both metrics are measured using the THOP library[1]. For FLOPs calculation, we use an input audio with a duration of **30 seconds**.

The results show that as $N$ increases, the model parameters increase linearly, but the increment between adjacent $N$ values is relatively small (about 0.033M parameters). The FLOPs for different $N$ are also very close, indicating that enlarging $N$ does not introduce significant extra computational cost. This suggests that increasing the number of voters achieves better performance and robustness with only minimal impact on model size and inference efficiency.

Table 12: Model parameters and inference FLOPs for different voter counts $N$ on a 30-second input audio.

| Number of Voters ($N$) | #Parameters (M) | #FLOPs (G) |
|---|---|---|
| $N = 1$ | 320.261 | 480.978 |
| $N = 3$ | 320.294 (↑ 0.010%) | 481.003 (↑ 0.005%) |
| $N = 5$ | 320.328 (↑ 0.021%) | 481.028 (↑ 0.010%) |
| $N = 7$ | 320.361 (↑ 0.031%) | 481.053 (↑ 0.016%) |

### B.6.2 EMPIRICAL INFERENCE EFFICIENCY BENCHMARK

From an architectural standpoint, the overhead introduced by the multi-branch design is negligible. The core design of StableToken confines the multi-branch architecture to the lightweight quantization stage only, while the computationally expensive Transformer Encoder remains a single-path process. Since the parallel quantization branches can be executed concurrently with high efficiency on modern hardware, this design introduces almost no additional end-to-end latency compared to conventional single-path tokenizers.

To rigorously evaluate the practical inference efficiency of StableToken, we conducted a comprehensive benchmark test measuring latency, Real-Time Factor (RTF), throughput, and memory footprint. We compared StableToken against a strong baseline, GLM-4-Voice-Tokenizer (Zeng et al., 2025), under identical hardware conditions using both an NVIDIA H20 GPU and an AMD EPYC 9K84 96-Core Processor.

The tests covered various batch sizes and audio durations. The detailed results are summarized in Table 13 (GPU) and Table 14 (CPU).

Based on these results, we observe the following: (1) **Latency, RTF, and Throughput:** The empirical data aligns with our theoretical analysis. The end-to-end latency and RTF of StableToken are nearly identical to the baseline, showing a slight advantage at larger batch sizes (e.g., 2.0% faster at a batch

---

[1] https://github.com/Lyken17/pytorch-OpCounter

Table 13: Inference Efficiency and Resource Usage on a single NVIDIA H20 GPU.

| Batch Size | Duration (s) | Metric | StableToken (25 tokens/s) | GLM-4-Voice (12.5 tokens/s) | Difference (%) |
|---|---|---|---|---|---|
| 1 | 1 | Latency (ms) | 9.11 | 9.00 | +1.22% |
| | | RTF | 0.0091 | 0.0090 | +1.22% |
| | | Throughput (s/s) | 109.77 | 111.11 | -1.21% |
| | | Memory (MB) | **1330** | 1538 | **-13.5%** |
| 1 | 30 | Latency (ms) | 63.17 | 63.19 | -0.03% |
| | | RTF | 0.0021 | 0.0021 | -0.03% |
| | | Throughput (s/s) | 474.92 | 474.79 | +0.03% |
| | | Memory (MB) | **1693** | 1853 | **-8.6%** |
| 16 | 10 | Latency (ms) | 235.96 | 239.57 | -1.5% |
| | | RTF | 0.0015 | 0.0015 | -1.5% |
| | | Throughput (s/s) | 678.08 | 667.87 | +1.5% |
| | | Memory (MB) | **2173** | 2328 | **-6.7%** |
| 32 | 60 | Latency (ms) | **1622.86** | 1656.49 | **-2.0%** |
| | | RTF | **0.0008** | 0.0009 | **-2.0%** |
| | | Throughput (s/s) | **1183.10** | 1159.08 | **+2.1%** |
| | | Memory (MB) | 13978 | 14230 | -1.8% |

Table 14: Inference Efficiency and Resource Usage on an AMD EPYC 9K84 96-Core Processor.

| Batch Size | Duration (s) | Metric | StableToken (25 tokens/s) | GLM-4-Voice (12.5 tokens/s) | Difference (%) |
|---|---|---|---|---|---|
| 1 | 1 | Latency (ms) | **104.27** | 116.62 | **-10.6%** |
| | | RTF | **0.1043** | 0.1166 | **-10.6%** |
| | | Throughput (s/s) | **9.59** | 8.57 | **+11.9%** |
| | | Memory (MB) | **2349** | 2727 | **-13.9%** |
| 1 | 30 | Latency (ms) | 1790.86 | 1809.86 | -1.0% |
| | | RTF | 0.0597 | 0.0603 | -1.0% |
| | | Throughput (s/s) | 16.75 | 16.58 | +1.0% |
| | | Memory (MB) | **2357** | 2739 | **-13.9%** |
| 16 | 10 | Latency (ms) | **5287.85** | 5687.53 | **-7.0%** |
| | | RTF | **0.0330** | 0.0355 | **-7.0%** |
| | | Throughput (s/s) | **30.26** | 28.13 | **+7.6%** |
| | | Memory (MB) | **2566** | 2982 | **-14.0%** |
| 32 | 60 | Latency (ms) | **52021.89** | 55549.44 | **-6.4%** |
| | | RTF | **0.0271** | 0.0289 | **-6.4%** |
| | | Throughput (s/s) | **36.91** | 34.56 | **+6.8%** |
| | | Memory (MB) | **2957** | 3117 | **-5.1%** |

size of 32 on GPU). Throughput matches or slightly exceeds the baseline across all tested scenarios. (2) **Memory Footprint:** StableToken consistently consumes less memory than the baseline, with savings of up to 13.5% on GPU and 14.0% on CPU. We attribute this efficiency to the use of Lookup-Free Quantization (LFQ), which is inherently more resource-efficient than the Vector Quantization (VQ) employed by the baseline.

In summary, the benchmarks demonstrate that StableToken achieves improved token stability without compromising inference efficiency. Its latency and throughput are competitive with strong baselines, while offering a lower memory footprint.

## B.7 ANALYSIS OF TOKEN DISTRIBUTION AND CROSS-LINGUAL EFFICIENCY

In this section, we investigate the fundamental properties of the token distribution generated by our multi-branch architecture. Specifically, we analyze the entropy, vocabulary efficiency, and language-specificity of the learned representations. We conduct a statistical analysis on the Chinese and English token distributions to verify whether the architecture effectively learns distinct phonetic representations without introducing significant bias or inefficiency.

Our analysis focuses on two main aspects: (1) Evaluating the efficiency of vocabulary space utilization for individual languages. (2) Assessing the degree of distributional overlap between Chinese and English to confirm the capture of language-specific features.

### B.7.1 VOCABULARY UTILIZATION AND DISTRIBUTION ENTROPY

We first analyze the vocabulary usage of StableToken on the LibriSpeech-test (English) and AISHELL (Chinese) datasets. As shown in Table 15, both languages utilize a substantial portion of the available vocabulary. The high entropy values observed for both languages indicate that the token distributions are rich and dispersed, rather than collapsing into a small subset of tokens. This suggests that the tokenizer maintains high representational capacity for both languages.

Table 15: Vocabulary Utilization Statistics on LibriSpeech and AISHELL

| Metric | English | Chinese |
|---|---|---|
| Total Tokens | 969,440 | 906,250 |
| Unique Tokens Used | 8,139 (99.35%) | 7,565 (92.35%) |
| Token Entropy | 12.43 | 11.68 |

### B.7.2 CROSS-LINGUAL TOKEN SPECIFICITY

To assess the language specificity of the learned codebook, we examine the overlap of high-frequency tokens between languages. We identify the top-frequency tokens for one language and calculate their occurrence frequency in the other.

Tables 16 and 17 present these comparisons. The results indicate that tokens with high frequency in one language are extremely rare or non-existent in the other. This disjointed usage pattern demonstrates that the model has successfully learned specialized, language-specific phonetic units, allocating distinct sub-spaces of the codebook to different languages.

Table 16: Frequency Analysis of Top 5 English Tokens

| Token ID | Freq. (English) | Freq. (Chinese) | Rank in Chinese |
|---|---|---|---|
| 1877 | 0.4803% | 0.0119% | 2455 |
| 3813 | 0.3693% | **0.00%** | N/A |
| 3809 | 0.3377% | **0.00%** | N/A |
| 3812 | 0.3090% | **0.00%** | N/A |
| 3808 | 0.2957% | 0.0001% | 7320 |

Table 17: Frequency Analysis of Top 5 Chinese Tokens

| Token ID | Freq. (Chinese) | Freq. (English) | Rank in English |
|---|---|---|---|
| 3810 | 0.7531% | 0.0042% | 6276 |
| 3811 | 0.7530% | 0.0002% | 8045 |
| 7910 | 0.7392% | 0.0027% | 6850 |
| 3815 | 0.6933% | 0.0001% | 8104 |
| 3234 | 0.6815% | 0.0001% | 8085 |

### B.7.3 QUANTITATIVE DISTRIBUTIONAL DIVERGENCE

To formally quantify the difference between the token distributions of the two languages, we calculate the Kullback-Leibler (KL) Divergence. As shown in Table 18, the significant non-zero KL Divergence values confirm that the model maintains distinct probability distributions for each language.

Table 18: KL Divergence between Language Token Distributions

| Metric | Value |
| --- | --- |
| KL Divergence (English ‖ Chinese) | **2.81** |
| KL Divergence (Chinese ‖ English) | **2.05** |

Collectively, these analyses demonstrate that the multi-branch voting mechanism robustly allocates specific regions of the codebook to different languages, capturing unique phonetic characteristics while maintaining efficient vocabulary utilization.

## B.8 LONG AUDIO PROCESSING AND BOUNDARY STABILITY

In this section, we clarify the model's configuration regarding input duration and analyze the stability of tokenization at segment boundaries. This analysis is crucial for understanding the model's behavior in streaming scenarios or when processing long audio via chunking.

### B.8.1 MODEL CONFIGURATION AND SEGMENT LENGTH

StableToken inherits the architectural specifications and processing conventions of its backbone, `whisper-large-v3`. Consequently, the model is configured as follows: (1) **Maximum Duration:** The tokenizer processes a maximum input audio duration of 30 seconds per inference pass, consistent with the Whisper context window. (2) **Training Constraint:** During training, all audio segments were constrained to this 30-second window. Audio files exceeding this duration in the training dataset were truncated to the first 30 seconds. (3) **Inference Strategy:** For audio exceeding 30 seconds, StableToken employs a standard chunking strategy. The input is segmented into 30-second (or smaller) chunks which are processed independently. The resulting token sequences are then concatenated to form the final output.

### B.8.2 ANALYSIS OF BOUNDARY STABILITY

A potential concern with chunk-based processing is whether token stability degrades at the boundaries of audio segments (i.e., the beginning and end of a chunk), which could lead to defects when chunks are concatenated together.

To investigate this, we analyzed the distribution of tokenization inconsistencies across different temporal regions of the input audio under Gaussian noise perturbation. We segmented the test audio clips into three regions: (1) **Start:** The first 15% of the audio duration; (2) **Middle:** The central 70% of the audio duration; and (3) **End:** The final 15% of the audio duration. We calculated the Unit Edit Distance (UED) independently for each region. The results are presented in Table 19.

Table 19: Distribution of UED across different temporal regions of audio clips under Gaussian noise.

| Region | Inconsistencies | Ref. Tokens | UED | % of Total Inconsistencies |
| --- | --- | --- | --- | --- |
| Start (0-15%) | 16,585 | 119,770 | 13.85% | 16.29% |
| Middle (15-85%) | 69,333 | 550,069 | 12.60% | 68.11% |
| End (85-100%) | 15,884 | 116,016 | 13.69% | 15.60% |
| **Overall** | 101,802 | 785,855 | 12.95% | 100.00% |

The empirical results indicate that UED is consistent across all regions. The UED in the middle region (12.60%) is only marginally lower than that at the start (13.85%) and end (13.69%). Furthermore, the

proportion of total inconsistencies occurring in the start (16.29%) and end (15.60%) regions closely aligns with their respective 15% share of the audio duration.

This uniformity demonstrates that there is no significant degradation in token stability at the segment boundaries. The consensus mechanism employed by StableToken maintains robustness consistently throughout the audio clip, supporting the effectiveness of the chunking strategy for long audio processing.

## C  ABLATION STUDY ON ARCHITECTURAL HYPERPARAMETERS

In this section, we investigate the impact of two key architectural hyperparameters on model performance: the depth at which the quantizer is placed within the encoder, and the ratio of clean to perturbed branches during training. We compare our proposed configuration (**L16 + 3:2**, which means the quantizer is placed at Layer 16 with a 3:2 clean-to-noisy branch ratio) against three variants: (1) **Shallower Quantization (L8 + 3:2)**: The quantizer is placed at Layer 8; (2) **Deeper Quantization (L24 + 3:2)**: The quantizer is placed at Layer 24; and (3) **Reduced Noise Ratio (L16 + 4:1)**: The ratio of clean to perturbed branches is set to 4:1.

Table 20 summarizes the performance of each model checkpoint at 100k training steps across tokenizer robustness (UED), Automatic Speech Recognition (ASR), and Speech Emotion Recognition (SER) tasks.

Table 20: Ablation study on quantizer placement and perturbation ratio. UED is calculated under Gaussian Noise. LibriSpeech WER is reported for test-clean / test-other. SER is evaluated on the ESD dataset.

| Configuration | Robustness | ASR (WER%, ↓) | | | SER |
|---|---|---|---|---|---|
| (Layer + Clean:Noisy) | (UED%, ↓) | LibriSpeech | WenetSpeech | KeSpeech | (Acc%, ↑) |
| L16 + 3:2 (Ours) | 18.68 | **2.22 / 5.38** | 10.91 | **11.00** | **67.38** |
| L8 + 3:2 | 22.05 | 2.39 / 5.82 | 11.84 | 11.24 | 66.90 |
| L24 + 3:2 | **13.65** | 2.52 / 5.96 | **10.16** | 11.03 | 59.51 |
| L16 + 4:1 | 20.94 | 2.40 / 5.91 | 11.00 | 11.57 | 62.14 |

### C.1  EFFECT OF QUANTIZER PLACEMENT

The results demonstrate a clear trade-off associated with the quantizer's depth: (1) **Deep Quantization (L24):** Placing the quantizer at the deep layer forces it to operate on highly abstract, semantic features. This yields the highest tokenizer robustness (lowest UED) and strong ASR performance on challenging datasets like WenetSpeech. However, at this depth, much of the fine-grained acoustic information (e.g., prosody and timbre) has been abstracted away, resulting in a significant degradation in SER performance (59.51% vs. 67.38%). (2) **Shallow Quantization (L8):** Operating on shallower features at Layer 8 retains acoustic details but lacks sufficient semantic abstraction. This leads to both poorer tokenizer robustness and degraded ASR performance compared to the deeper configurations. (3) **Balanced Configuration (L16):** Our proposed placement at Layer 16 strikes an effective balance. It captures sufficiently robust semantic content for high-performance ASR while retaining enough acoustic detail to excel in paralinguistic tasks like SER.

### C.2  EFFECT OF PERTURBATION RATIO

Comparing the 3:2 and 4:1 clean-to-noisy ratios reveals the importance of the training signal difficulty: The 3:2 ratio results in significantly better tokenizer robustness compared to the 4:1 ratio (18.68% vs. 20.94% UED). A higher proportion of perturbed branches creates a more challenging training objective for the consensus mechanism.

This enhanced robustness translates directly to downstream tasks. We hypothesize that the more diverse training signal forces the model to learn representations that are truly invariant to perturbations rather than relying on the clean majority. This richer representation benefits both semantic (ASR) and paralinguistic (SER) performance.

In conclusion, these ablation studies validate the design choices of the proposed model. The combination of Layer 16 placement and a 3:2 noise ratio optimizes the trade-off between semantic robustness and acoustic feature preservation.

## D  BASELINE MODELS

The baseline models considered in our study are as follows: (1) **HuBERT-500** (Hsu et al., 2021), a self-supervised speech representation model that leverages iterative offline clustering to produce pseudo-labels and employs a masked prediction loss; we use the official checkpoint with 500 clusters[2]; (2) **NAST** (Messica & Adi, 2024), a noise-aware speech tokenization approach comprising a predictor, residual encoder, and decoder, in which the predictor representations of clean speech and augmented speech are explicitly aligned; (3) **R-Spin** (Chang & Glass, 2024), which enhances the robustness of speech representations by learning discrete, speaker- and noise-invariant acoustic units through a prediction-based training objective; (4) **SpeechTokenizer** (Zhang et al., 2023), which introduces a hierarchical encoder-decoder framework with residual vector quantization (RVQ) to unify semantic and acoustic tokens; (5) **X-Codec** (Ye et al., 2025), which augments the RVQ backbone with semantic features from a pre-trained semantic encoder and applies a semantic reconstruction loss to achieve higher fidelity in audio generation; (6) **Mimi** (Défossez et al., 2024), a neural audio codec using RVQ to convert audio into discrete tokens, where the first quantization level is distilled to capture semantic information; (7) **CosyVoice ($\mathcal{S}^3$ Tokenizer)** (Du et al., 2024a), which extracts supervised semantic tokens from a multilingual speech recognition encoder for LLM-based TTS, thereby improving content consistency and speaker similarity in voice cloning; (8) **CosyVoice2** (Du et al., 2024b), which introduces finite-scalar quantization (FSQ) to improve the codebook utilization and updates the model architecture for streaming synthesis capabilities; (9) **GLM-4-Voice** (Zeng et al., 2025), which fine-tunes a pre-trained ASR model by including a pooling layer and a vector quantization layer, producing discrete tokens that strongly preserve semantic information at low frame rates.

## E  NOISE PROFILES

In tokenizer-level evaluation, we augment the FLEURS (Conneau et al., 2023) benchmark with a variety of synthetic perturbations, including Gaussian Noise, Pink Noise, Brown Noise and Bit Crush Distortion, as well as real-world noise samples from the ESC-50 (Piczak, 2015) dataset. Specifically, the ESC-10 (Piczak, 2015) subset is used as out-of-domain (OOD) real-world noise and is excluded from our StableToken training pipeline, while the remaining 40 noise categories from ESC-50 are incorporated into the training process and thus are considered in-domain real-world noise.

We carefully adjusted the noise level to ensure that the added noise does not obscure the semantic content of the original audio and does not affect human perception of the speech. A summary of all perturbation types and their corresponding intensity is provided in Table 21.

Table 21: Details of synthetic and real-world perturbations used for noise augmentation.

| Perturbation Type | Intensity Value |
|---|---|
| Gaussian Noise | SNR = 25 |
| Pink Noise | SNR = 22 |
| Brown Noise | SNR = 16 |
| Bit Crush Distortion | Bit Depth = 10 |
| Real-world Noise | SNR = 16 |
| OOD Real-world Noise | SNR = 16 |

---

[2]`https://github.com/facebookresearch/fairseq/tree/main/examples/hubert`

## F    DETAILS OF DOWNSTREAM TASK EVALUATION

### F.1    TRAINING DATASETS FOR SPEECHLLMS

In this section, we summarize the speech datasets employed for training SpeechLLMs in Table 22, covering various tasks including Automatic Speech Recognition (ASR), Speech Emotion Recognition (SER), Text-to-Speech (TTS), and Speech Next Token Prediction (SNTP).

Table 22: Summary of datasets used for training SpeechLLMs

| Dataset | Task | Language(s) |
|---|---|---|
| LibriSpeech (Panayotov et al., 2015) | ASR | English |
| Multi-Lingual Librispeech (Pratap et al., 2020) | ASR | English |
| TESS (Dupuis & Pichora-Fuller, 2010) | SER | English |
| SAVEE (Jackson & Haq, 2014) | SER | English |
| RAVDESS (Livingstone & Russo, 2018) | SER | English |
| MELD (Poria et al., 2018) | SER | English |
| MEAD (Wang et al., 2020) | SER | English |
| JL-Corpus (James et al., 2018) | SER | English |
| IEMOCAP (Busso et al., 2008) | SER | English |
| Expresso (Nguyen et al., 2023) | SER | English |
| EmoV-DB (Adigwe et al., 2018) | SER | English |
| EMNS (Noriy et al., 2023) | SER | English |
| Dailytalk (Lee et al., 2023) | SER | English |
| CREMA-D (Cao et al., 2014) | SER | English |
| CASIA (Zhang & Jia, 2008) | SER | Chinese |
| M3ED (Zhao et al., 2022) | SER | Chinese |
| MER2023 (Lian et al., 2023) | SER | Chinese |
| CSEMOTIONS (Tian et al., 2025) | SER | Chinese |
| ESD (Zhou et al., 2022) | SER | English, Chinese |
| Hi-Fi TTS (Bakhturina et al., 2021) | TTS | English |
| VCTK (Veaux et al., 2017) | TTS | English |
| LibriTTS (Zen et al., 2019) | TTS | English |
| GigaSpeech (Chen et al., 2021) | SNTP | English |
| VoxPopuli (Wang et al., 2021) | SNTP | English |
| MagicData[3] | TTS | Chinese |
| AISHELL-1 (Bu et al., 2017) | TTS | Chinese |
| WenetSpeech (Zhang et al., 2022) | SNTP | Chinese |

### F.2    TRAINING HYPERPARAMETERS FOR SPEECHLLMS

Table 23 summarizes the main hyperparameters used throughout downstream SpeechLLM training. Unless otherwise specified, these settings are uniformly adopted for all tasks and models.

### F.3    PROMPTS USED IN DOWNSTREAM TASKS

This appendix contains the complete lists of textual prompts used for Automatic Speech Recognition (ASR), Speech Emotion Recognition (SER), and Text-to-Speech (TTS) tasks. For both fine-tuning and inference, a prompt was randomly selected from the corresponding set for each sample.

---

[3] https://www.openslr.org/68/

Table 23: Hyperparameters used for training all downstream SpeechLLMs.

| Hyperparameter | Value |
| --- | --- |
| optimizer_type | Adam |
| lr_scheduler | Cosine |
| learning_rate | 4.0e-4 |
| min_lr | 4.0e-5 |
| lr_decay_ratio | 0.75 |
| weight_decay | 0.1 |
| grad_clip | 1.0 |

### F.3.1 ASR TASK PROMPTS

```
Please transcribe the following audio content into text.
Please convert the following recording into text.
Please transcribe this audio recording into text.
Transcribe the following audio content into text.
Convert the following recording into text.
This audio recording needs to be transcribed into text.
Recognize and convert the following speech content into text.
Turn the following audio file into text.
Transcribe this recording into text.
Transcribe the following audio file into text.
Convert this speech recording into text.
Recognize the following audio content and convert it into text.
Transcribe the following recording into text.
```

### F.3.2 SER TASK PROMPTS

```
What is the emotion of this text?
Analyze the sentiment of the following sentence.
Identify the feeling expressed in this audio.
Is the tone of this message positive or negative?
Detect the emotion in the user's feedback.
What emotion is being conveyed here?
Classify the emotion of this statement.
Tell me the emotional state of the speaker.
Analyze the emotional content of this speech.
```

### F.3.3 TTS TASK PROMPTS

```
Please synthesize the following text into speech.
Convert the following text to speech.
Transform the following text into speech.
This text needs to be synthesized into speech.
Synthesize the following text into speech.
Turn the following text into speech.
Generate speech from the following text.
Convert the text below into speech.
Create speech from the following text.
Produce speech from the following text.
Render the following text as speech.
```

# G  FULL RECONSTRUCTION RESULTS

The comprehensive results for the tokenizer-level reconstruction quality evaluation are provided in Table 24. Note that SSL-based semantic tokenizers are not included in this comparison, as there are no publicly available decoders for reconstructing audio from their generated tokens.

Table 24: WER (↓) and MOS (↑) on LibriSpeech (Panayotov et al., 2015) and SEED (Anastassiou et al., 2024). StableToken combines strong noise robustness with competitive reconstruction quality. It is worth noting that a comparison is most meaningful between tokenizers of the same type.

| Model | #C | Frame Rate | BPS | WER ↓ | | | | MOS ↑ | | | |
| --- | --- | --- | --- | --- | --- | --- | --- | --- | --- | --- | --- |
| | | | | LS-clean | LS-other | SEED en | SEED zh | LS-clean | LS-other | SEED en | SEED zh |
| **Semantic Distilled tokenizer** | | | | | | | | | | | |
| SpeechTokenizer (Zhang et al., 2023) | 1 | 50Hz | 500 | 4.77 | 16.06 | 10.37 | 74.98 | 2.51 | 2.49 | 2.51 | 2.44 |
| | 3 | 50Hz | 1500 | 4.03 | 10.72 | 4.93 | 7.81 | 3.00 | 2.89 | 2.89 | 3.06 |
| | 8 | 50Hz | 4000 | 3.21 | 6.58 | 2.77 | 2.25 | 3.32 | 3.10 | **3.22** | **3.44** |
| X-Codec (Ye et al., 2025) | 1 | 50Hz | 500 | 3.98 | 9.02 | 4.72 | 5.96 | 3.17 | 3.04 | 3.05 | 3.18 |
| | 3 | 50Hz | 1500 | 3.16 | 6.11 | 2.74 | 2.24 | 3.43 | 3.17 | 3.19 | 3.38 |
| | 8 | 50Hz | 4000 | **3.09** | **5.49** | **2.25** | **1.74** | **3.47** | **3.19** | 3.19 | 3.33 |
| Mimi (Défossez et al., 2024) | 8 | 12.5Hz | 1100 | 4.65 | 9.84 | 3.86 | 2.81 | 3.26 | 3.06 | 3.15 | 3.19 |
| **Supervised Semantic tokenizer** | | | | | | | | | | | |
| GLM-4-Voice-Token. (Zeng et al., 2025) | 1 | 12.5Hz | 175 | 4.04 | 9.33 | 3.54 | 3.23 | 4.07 | **3.99** | **4.16** | 4.10 |
| $\mathcal{S}^3$ Tokenizer (Du et al., 2024a) | 1 | 25Hz | 300 | 5.78 | 13.38 | 5.91 | 4.26 | 3.40 | 3.31 | 3.40 | 3.31 |
| CosyVoice2 (Du et al., 2024b) | 1 | 25Hz | 325 | 4.25 | 9.68 | 4.34 | 2.75 | 3.36 | 3.25 | 3.31 | 3.58 |
| **StableToken (Ours)** | 1 | 25Hz | 325 | **3.84** | **7.99** | **3.44** | **2.62** | **4.09** | 3.83 | 4.01 | **4.18** |

