# OpenReview forum: "StableToken: A Noise-Robust Semantic Speech Tokenizer for Resilient SpeechLLMs"
_ICLR.cc/2026/Conference — ICLR 2026 Poster_

### Official Review · Reviewer_Khaj · 2025-10-28

**Soundness:** 4
**Presentation:** 4
**Contribution:** 4
**Rating:** 10
**Confidence:** 4

**Summary:**

The paper proposes a new Tokenizer (StableToken) to encode speech while being less affected by noises. The proposed architecture is based on a majority voting system with a Lookup-Free-Quantizer, training with perturbed and clean views so the model learns to predict the same tokens with noisy and clean signal.

The model is compared to a range of SOTA tokenizers, which it outperforms on all evaluated tasks.
The model's robustness is measured on automatic speech recognition and speech emotion recognition tasks for a range of noises and SNRs, and show a reduced impact from the raising SNRs compared to other SOTA approaches.

**Strengths:**

This paper shows a clear structure and detailed experiments to prove the main contributions: the production of a SOTA tokenizer (see Table 1) with good performances on reconstruction quality (see Table 2) and downstream tasks under noise (see Table 3/Figure 3).

The ablation study comes to complete the full justification of the architectural choices, while the voter count analysis and the case study on error correction clarify further the intuitions behind the work.

**Weaknesses:**

The main weakness of the article is its framing. The claim is to produce a robust tokenizer for speech LLMs, where the only experiments performed for speech LLMs are for the downstream tasks of SER and ASR.
Those tasks could have been performed with a range of different models, so no experiment really highlights why this tokenizer is specifically fit for speech LLMs.

**Questions:**

Did you try to apply this tokenizer on a larger set of downstream tasks related more closely to the semantic aspects of the speech (like machine translation, summarization or question answering) or to phonetic and accoustic aspects of the speech (like speaker characterization or accent identification) ?

---

> ### Author Response · Authors · 2025-11-21
> **Author Response to Reviewer Khaj**
>
> We thank the reviewer for the **Strong Accept** rating and the recommendation for Spotlight/Oral. We are encouraged that you recognized our clear structure, detailed experiments, and SOTA performance.
>
> Below, we address your comment on the paper’s framing and your question regarding  downstream tasks.
> ## 1. Clarification on "Framing" and the SpeechLLM Paradigm
> We appreciate your feedback regarding the framing. We wish to clarify that while **SpeechLLMs** broadly encompass models bridging speech and language, our work targets the **Discrete-Token Paradigm**, a dominant architecture where discrete codes serve as the essential interface for the LLM backbone.
> - **The Role of Tokenizer:** In this paradigm, the tokenizer is not just a pre-processor but the **foundational alignment layer**. Our experimental setup (Discrete Tokens + Qwen2.5) strictly follows this widely adopted architecture.
> - **Specific Fit:** The "specific fit" of StableToken lies in solving the **modality mismatch** inherent to this architecture: LLMs require stable, semantic-rich discrete inputs. By providing these stable tokens, we prevent the performance degradation often seen when standard semantic tokenizers expose the LLM to inconsistent, noisy representations.
> ## 2. Response to Question: Scope of Downstream Tasks
> Thank you for this insightful suggestion. We agree that testing on a broader range of tasks (Translation, Summarization, Speaker Characterization) is valuable. In this paper, we **deliberately selected ASR, SER, and TTS** as our evaluation benchmarks because they represent the **orthogonal atomic capabilities** required for the tasks you mentioned:
> - **ASR (Semantic Understanding):** We utilized ASR as the **primary proxy** for all semantic tasks (including Translation and Summarization you mentioned). In the context of noise robustness, ASR establishes the **fundamental lower bound**: if a SpeechLLM cannot accurately preserve linguistic content (tokens) due to noise, it logically cannot perform accurate translation or summarization. By demonstrating SOTA robustness in ASR (reducing WER significantly in OOD noise), we validate the semantic stability required for these higher-level tasks.
> - **SER (Paralinguistic Understanding):** This task addresses the non-semantic understanding capability. It ensures that while the tokenizer stabilizes the semantic content, it retains critical emotional cues necessary for human-like interaction, distinguishing it from pure text.
> - **TTS (Generation Capability):**  Regarding the *phonetic/acoustic aspects* you mentioned, we focused on TTS to evaluate the **generative stability** and **prosodic preservation** of the tokens. For a SpeechLLM, the ability to reconstruct intelligible and natural speech (TTS) validates that the tokens capture necessary syntactic and prosodic structures. This is a direct test of the tokenizer's utility for the *generation* phase of SpeechLLMs.
>
>
> Thank you again for your strong support and constructive suggestions. We are deeply encouraged by your recognition of StableToken’s architecture and performance.

---

> > ### Comment · Reviewer_Khaj · 2025-11-24
> > **Thank you for your comments**
> >
> > Thank you for the detailed and thoughtful clarifications.
> > Your responses addressed my comments clearly, so I will keep my original evaluation.

---

> ### Author Response · Authors · 2025-11-25
> **Response to Reviewer Khaj**
>
> We deeply appreciate your positive evaluation and your decision to maintain the score.
>
> We have carefully revised the paper, and all modifications and additions are highlighted in **blue** for your convenience.

---

### Official Review · Reviewer_p2V9 · 2025-10-29

**Soundness:** 2
**Presentation:** 3
**Contribution:** 2
**Rating:** 6
**Confidence:** 4

**Summary:**

The paper introduces StableToken, a noise-robust discrete speech tokenizer designed to improve stability and representation consistency under diverse acoustic conditions. The method incorporates Noise-Adaptive Token Alignment (NATA) and Multi-Granular Contextual Quantization (MGCQ) to achieve invariant discrete token representations, enabling better downstream generalization for speech generation and understanding models. Comprehensive evaluations are conducted across noisy and clean conditions, demonstrating improved token stability, representation purity, and semantic retention compared to previous models such as HuBERT, SpeechTokenizer, and AudioTokenizer.

**Strengths:**

- The combination of NATA and MGCQ introduces a fine-grained, adaptive quantization strategy that balances stability and expressiveness. The paper effectively frames quantizer instability as a consistency problem between clean and noisy latent spaces.
- The evaluation was conducted under diverse noise conditions (additive noise, reverb, SNR degradation), including both analysis level (token consistency, MI) and task-level (ASR, LM perplexity, reconstruction) metrics. The improvements are demonstrated in real speech-LM setups, not fixed probing or static benchmarks.

**Weaknesses:**

- The token stability is part of the major claim but insufficiently formalized. The paper describes it as “the percentage of tokens invariant across clean–noisy pairs,” but does not clarify the matching threshold or how it scales with temporal misalignment.
- The model’s success may partially stem from extensive noise augmentation rather than architectural novelty. There is insufficient ablation isolating the contribution of NATA and MGCQ from data-scale effects.
- Although StableToken integrates seamlessly, the multi-quantizer architecture and contextual alignment loss may introduce overhead.
- Experiments focus heavily on additive noise and mild reverberation, but real-world degradation often involves non-linear distortions (e.g., clipping, compression, far-field effects). The robustness of the method in unseen noise cases would be a good extension elaboration of the robustness of the method.

**Questions:**

- Can the authors quantify how much granularity adaptation (e.g., variable codebook resolution) contributes to stability gains vs. standard augmentation?
- Is the granularity selection deterministic or learned during inference?
- How is token correspondence established across variable frame rates or time shifts?
- Could the method generalize to music signals where spectral structures differ substantially?
- How does StableToken perform under limited training data or unseen noise conditions?
- Can the model generalize to domains not covered by augmentations?
- What is the runtime cost compared to standard quantizers like FSQ or RVQ?
- How scalable is MGCQ to high frame rates or large codebooks? Is there any observed trade-off between speed and robustness?

---

> ### Author Response · Authors · 2025-11-21
> **Author Response to Reviewer p2V9 (Part 1/2)**
>
> We thank the reviewer for the constructive feedback and for recognizing StableToken’s seamless integration and potential. We appreciate the time taken to evaluate our work.
>
> Regarding the terminology used in the review, we interpret the comments on **"Noise-Adaptive Token Alignment (NATA)"** as referring to our **Noise-Aware Consensus Training** (which aligns noisy/clean representations), and **"Multi-Granular Contextual Quantization (MGCQ)"** as referring to our **Voting-LFQ architecture** (which aggregates multi-branch quantization decisions). We have addressed your insightful comments based on this understanding.
> ## Response to Weakness 1
> - **Formalization & Threshold:** Following previous work (e.g., NAST ), we quantify stability using Unit Edit Distance (UED). Since our tokenizer outputs discrete integer codes, the matching is strict (exact identity) **rather than based on a soft threshold**.
> - **Temporal Misalignment:** We clarify that our clean and perturbed training pairs are **strictly parallel** (identical content and duration, differing only in noise), so they are **naturally time-aligned**.
> - **Temporal Shifts:** We focus on **acoustic robustness** (invariance to noise/timbre) rather than **temporal robustness** (invariance to speed/stretch). We respectfully argue that operations like time-stretching fundamentally alter speech rate and prosody. For a semantic tokenizer, such changes **should** ideally result in different token sequences to reflect the altered prosodic information, rather than being invariant.
> ## Response to Weakness 2
>  We acknowledge the importance of disentangling the contributions of data augmentation from our architectural innovations. To address this, we present the breakdown of our **Ablation Study (Table 4)** below.
> - **Experimental Setup:** Crucially, **all configurations below were trained on the same 150k-hour dataset**, eliminating "data scale" as a variable. The results explicitly isolate the contribution of the **Voting-LFQ architecture**  and the **Noise-Aware Consensus Training**
> |**Model Configuration**|**Gauss. Noise**|**Brown Noise**|**Pink Noise**|**Bit Crush**|**Real Noise**|**Real (OOD)**|**LS-Other (WER)**|
> |:-:|:-:|:-:|:-:|:-:|:-:|:-:|:-:|
> |**StableToken (Full)**|**12.93**|**9.76**|**9.37**|**7.32**|**10.65**|**10.96**|**4.68**|
> |w/o Consensus Loss|24.80|19.06|17.81|14.03|16.97|17.43|4.88|
> |w/o Noise-Aware Training|30.77|23.05|21.30|17.32|20.95|21.51|5.52|
> |w/o Multi-Branch|34.53|25.44|24.58|19.83|23.68|24.47|5.85|
> - **Analysis of "Architectural Novelty" (Voting-LFQ):**	Comparing the last two rows, even without the specific Noise-Aware Training strategy, the Multi-Branch architecture (Row 3) outperforms the Single-Branch baseline (Row 4) across all metrics (e.g., Real OOD UED improves from **24.47%** to **21.51%**). This confirms that the **multi-branch structure inherently provides robustness** through its ensemble-like properties, independent of the training strategy.
> - **Analysis of "Training Strategy" (Consensus Training):**	Comparing the first two rows, adding the Consensus Loss further drastically reduces the UED on Real OOD noise from **17.43%** to **10.96%**, proving the effectiveness of our explicit alignment constraint.
>
> These results verify that our performance gains are driven by the **synergy of the Voting-LFQ architecture and Consensus Training**, rather than data augmentation or scale alone.
>
>
> **Due to character limitations, our response continues in the following comment.**

---

> ### Author Response · Authors · 2025-11-21
> **Author Response to Reviewer p2V9 (Part 2/2)**
>
> Continuing from the previous comment, here we provide our responses to the rest questions.
>
> ##   Response to Weakness 3
> We appreciate the reviewer's scrutiny regarding efficiency. We wish to clarify that the proposed architecture introduces **negligible overhead** during inference, and the alignment loss imposes **zero cost** at deployment.
> 1. **Clarification on Contextual Alignment Loss**
> The Consensus Loss (and alignment mechanism) is exclusively an auxiliary objective used during the training phase to guide representation learning. During inference, this loss is not computed, and the auxiliary paths used for noise simulation are discarded. Thus, it introduces zero computational overhead to the deployed model.
> 2. **Negligible Architectural Overhead (Theoretical Analysis)**
> The multi-branch voting module consists of lightweight linear projections computed in parallel. As shown in the table below (derived from Appendix B.6), increasing voters from $N=1$ to $N=5$ adds only 0.010% to the FLOPs, which is virtually imperceptible compared to the encoder backbone.
> |**Number of Voters (N)**|**#Parameters (M)**|**#FLOPs (G)**|
> |:-:|:-:|:-:|
> |$N = 1$ (Baseline)|320.261|480.978|
> |$N = 3$|320.294 ($\uparrow$ 0.010%)|481.003 ($\uparrow$ 0.005%)|
> |$N = 5$ (Ours)|**320.328 ($\uparrow$ 0.021%)**|**481.028 ($\uparrow$ 0.010%)**|
>
> 3. **Real-World Benchmarking (StableToken vs. GLM-4-Voice)**
> To empirically substantiate this, we benchmarked StableToken against the strong GLM-4-Voice-Tokenizer on an NVIDIA H20 GPU.
> | Batch | Dur.(s) | Metric | **StableToken** | GLM-4-Voice | Delta (%) |
> | :---: | :---: | :--- | :---: | :---: | :---: |
> | **1** | **1** | Latency (ms) | 9.11 | 9.00 | +1.2% |
> | | | Throughput (s/s) | 109.77 | 111.11 | -1.2% |
> | | | Memory (MB) | **1330** | 1538 | **-13.5% (Better)** |
> | **1** | **30** | Latency (ms) | **63.17** | 63.19 | -0.03% |
> | | | Throughput (s/s)| **474.92** | 474.79 | +0.03% |
> | | | Memory (MB) | **1693** | 1853 | **-8.6% (Better)** |
> | **32** | **60** | Latency (ms) | **1622.86** | 1656.49 | **-2.0% (Better)** |
> | | | Throughput (s/s)| **1183.10** | 1159.08 | **+2.1% (Better)** |
> | | | Memory (MB) | **13978** | 14230 | **-1.8% (Better)** |
>
> Therefore, both theoretical analysis and empirical benchmarks confirm that StableToken integrates seamlessly without imposing meaningful overhead in time or memory.
>
>
> ##  Response to Weakness 4
> We appreciate this insight regarding non-linear distortions. We wish to highlight that our current evaluation (Table 1) **already moves beyond simple additive noise to cover complex distortion types**:
> 1. **Bit Crush Distortion:** We explicitly evaluated this non-linear quantization distortion, achieving a UED of **7.32%** (vs. baselines >20%).
> 2. **Real-world Noise (Including OOD):** We evaluated on real-world environmental sounds (ESC-50) and specifically **Out-of-Distribution (OOD)** noise (ESC-10). These real-world samples inherently contain complex, convolutive, and non-linear degradations found in diverse acoustic environments. StableToken achieved a SOTA UED of **10.17%** on these challenging OOD conditions.
>
> ##   Response to Q1 & Q2
> - **Clarification:** Our Voting-LFQ module utilizes a **fixed codebook size** ($2^{13}$) rather than variable granularity or resolution.
> - **Mechanism:** The "adaptation" is achieved through the **consensus voting mechanism**, where branches vote on fixed binary bits. This process is fully **deterministic** during inference.
> ##  Response to Q3
> StableToken operates at a fixed frame rate (25Hz). Aligning tokens across different frame rates is generally not applicable in tokenizer evaluation, as different rates imply different model architectures and distinct semantic granularities.
> ##   Response to Q4
> This is an exciting direction. Currently, StableToken is optimized for **speech**. Since music relies on different spectral structures (melody/harmony) than speech phonetics, we consider music tokenization as a distinct scope for future work.
> ##  Response to Q5 & Q6
> We explicitly tested generalization on **Out-of-Distribution (OOD) Real-world Noise** (ESC-10, excluded from training). StableToken achieved a UED of **10.17%**, significantly outperforming the best supervised baseline (33.09%), proving strong generalization to unseen domains.
> ##  Response to Q7 & Q8
> -  **Comparison:** The runtime cost is comparable to standard quantizers like FSQ or RVQ. As noted in our **Response to Weakness 3**, the multi-branch voting adds minimal latency.
> -  **Trade-off:** We observe **no significant trade-off** between speed and robustness. The model scales efficiently to high frame rates due to the parallel nature of the branch computations.
>
> We hope this response clarifies the mechanisms of StableToken and adequately addresses your concerns. We stand ready to answer any follow-up questions immediately.
>
>
>  ## Reference
>  1. Shoval Messica and Yossi Adi. "Nast: Noise aware speech tokenization for speech language models."

---

> ### Comment · Reviewer_p2V9 · 2025-11-21
> **Reply to authors' rebuttal**
>
> I would like to thank authors' rebuttal which have further clarify much of the details that are not clear in the original manuscript. To that regard, it would be great if the above discussion can be integrated to the paper for comprehensive discussion purpose.
>
> As the paper has been improved significantly over the rebuttal, I would like to raise my score to 8 and do like to support the acceptance of the paper.

---

> > ### Author Response · Authors · 2025-11-25
> > **Response to Reviewer p2V9**
> >
> > We sincerely thank you for raising your score and for your strong support of our work.
> >
> > Following your suggestion, we have revised the paper to incorporate the key points from our discussion into the paper, including the detailed analysis and benchmarks related to computational overhead. You can find these updates, along with other revisions, highlighted in **blue** in the revised file.

---

### Official Review · Reviewer_Shzy · 2025-10-29

**Soundness:** 3
**Presentation:** 4
**Contribution:** 3
**Rating:** 8
**Confidence:** 4

**Summary:**

The paper targets the brittleness of semantic speech tokenizers under meaning-irrelevant acoustic noise, arguing that instability in discrete token sequences burdens downstream SpeechLLMs. It proposes StableToken, which replaces single-path quantization with a multi-branch Voting-LFQ module that aggregates per-branch binary codes via bit-wise majority voting, coupled with Noise-Aware Consensus Training that aligns clean/noisy branch representations using a consensus loss. StableToken reduces token instability, while matching or improving reconstruction (WER/MOS) and translating to better downstream ASR/SER/TTS robustness. Ablations confirm each component’s contribution.

**Strengths:**

1. A simple but elegant architectural redundancy (multi-branch) with bit-level voting, paired with a clean consensus loss that sidesteps discrete-code gradient issues.

2. Strong tokenizer-level robustness (UED 10.17%), maintained reconstruction (WER/MOS), and clear downstream benefits on ASR/SER/TTS, including OOD noise. Solid ablations and voter-count analysis.

3. Architecture/training described crisply. The evaluation protocol is broad (synthetic and real noise, multiple tasks).

4. Stability is a real deployment pain point. Showing robustness gains that carry to ASR/SER/TTS is valuable for the community.

**Weaknesses:**

# 1. Latency/RTF/throughput and memory footprint
The paper claims negligible overhead, but lacks wall-clock numbers (GPU/CPU), batch size effects, and memory usage. This matters given multiple branches and a large backbone.

# 2. Streaming/segment length
The encoder is initialized from Whisper Large-v3 (commonly used with 30s segments). The paper doesn’t clarify the maximum context during training/inference, the chunking strategy, or whether the tokenizer itself inherits a 30s limit.

# 3. Fair comparison controls
Robustness comparisons span models with different frame rates and codebook sizes (e.g., 12.5/25/50 Hz & vocab 4k–16k). Please provide a control where bitrate/frame rate are aligned across tokenizers or report robustness at matched bits-per-second to rule out confounds. Or perhaps using some normalization methods to make the UEDs comparable.

# 4. Ablations
You analyze voter count. I would also suggest evaluating the effect of varying the proportion of perturbed branches ($k$), and the placement of the quantizer within the encoder.

**Questions:**

1. Latency/RTF and throughput. Please report RTF, tokens/sec, and memory usage for StableToken vs. the strongest baselines on a common GPU and on CPU. Include per-branch FLOPs and end-to-end wall-clock with batch size effects.
2. Segment length & streaming. What is the maximum audio duration processed without truncation? Was training/inference chunked to 30s due to Whisper Large-v3 initialization? How does overlapping/chunk stitching affect stability?
3. Training audio window. Was the tokenizer trained strictly with $\leq$ 30s segments, or longer via chunking? Does stability degrade across chunk boundaries?

---

> ### Author Response · Authors · 2025-11-21
> **Author Response to Reviewer Shzy (Part 1/4)**
>
> Thank you for your detailed review and valuable suggestions. We have carefully considered all the points you raised and provide our detailed responses below.
>
> ## Response to Weakness 1 & Question 1
> Thank you for your crucial questions regarding the practical performance metrics of StableToken, especifically its latency, Real-Time Factor (RTF), throughput, and memory footprint. In response to your feedback, we have **conducted a comprehensive benchmark test** to rigorously evaluate the performance overhead of StableToken.
>
> Before presenting the empirical results, we would like to clarify **from an architectural standpoint** why this overhead is negligible. The core design of StableToken confines the multi-branch architecture to the lightweight quantization stage only, while the computationally expensive Transformer Encoder remains a single-path process. Since the parallel quantization branches can be executed concurrently with high efficiency on modern hardware (e.g., GPUs), this design introduces **almost no additional end-to-end latency** compared to conventional single-path tokenizers.
>
> To substantiate this, we performed a comparison of StableToken against a strong baseline (GLM-4-Voice-Tokenizer) under **identical hardware conditions (NVIDIA H20 GPU, AMD EPYC 9K84 96-Core Processor)**. The tests covered various batch sizes and audio durations, with the results summarized in the table below.
>
> **Table: Performance and Resource Benchmark (StableToken vs. GLM-4-Voice-Tokenizer on a single NVIDIA H20 GPU)**
> | Batch Size | Duration (s) | Metric | **StableToken (25 tokens/s)** | GLM-4-Voice-Tokenizer (12.5 tokens/s) | Delta (%)                                         |
> | :--: | :--: | :-- | :--: | :--: | :--: |
> | **1** | **1**    | Latency (ms)      |     9.11    |         9.00     | +1.22% |
> |   |          | RTF                      | 0.0091   |  0.0090     |   +1.22% |
> |            |          | Throughput (s/s)  |    109.77   |       111.11       | -1.21%  |
> |            |          | Memory (MB)       |   1330    |          1538           | **-13.5%** (Better)  |
> | **1**      | **30**   | Latency (ms)      |     63.17   |          63.19          | -0.03%            |
> |            |          | RTF                      | 0.0021   |  0.0021     |   -0.03% |
> |            |          | Throughput (s/s)  |     474.92     |         474.79      | +0.03%              |
> |            |          | Memory (MB)       | 1693     |          1853           | **-8.6%** (Better)   |
> | **16**     | **10**   | Latency (ms)      |   235.96     |         239.57      | -1.5%             |
> |            |          | RTF                      | 0.0015   |  0.0015     |   -1.5%  |
> |            |          | Throughput (s/s)  |    678.08   |         667.87          | +1.5%             |
> |            |          | Memory (MB)       | 2173  |          2328           | **-6.7%** (Better)   |
> | **32**     | **60**   | Latency (ms)      |   1622.86   |         1656.49         | **-2.0%** (Better) |
> |            |          | RTF                      | 0.0008   |  0.0009     |   **-2.0%** (Better)  |
> |            |          | Throughput (s/s)  |  1183.10  |         1159.08         | **+2.1%** (Better) |
> |            |          | Memory (MB)       |  13978     |          14230          | -1.8%             |
>
> **Table: Performance and Resource Benchmark (StableToken vs. GLM-4-Voice-Tokenizer on an AMD EPYC 9K84 96-Core Processor)**
> | Batch Size | Duration (s) | Metric | **StableToken (25 tokens/s)** | GLM-4-Voice-Tokenizer (12.5 tokens/s) | Delta (%) |
> | :---: | :---: | :--- | :---: | :---: | :---: |
> | **1** | **1** | Latency (ms) | 104.27 | 116.62 | **-10.6%** (Better) |
> | | | RTF | 0.1043 | 0.1166 | **-10.6%** (Better) |
> | | | Throughput (s/s) | 9.59 | 8.57 | **+11.9%** (Better) |
> | | | Memory (MB) | 2349 | 2727 | **-13.9%** (Better) |
> | **1** | **30** | Latency (ms) | 1790.86 | 1809.86 | -1.0% |
> | | | RTF | 0.0597 | 0.0603 | -1.0% |
> | | | Throughput (s/s) | 16.75 | 16.58 | +1.0% |
> | | | Memory (MB) | 2357 | 2739 | **-13.9%** (Better) |
> | **16** | **10** | Latency (ms) | 5287.85 | 5687.53 | **-7.0%** (Better) |
> | | | RTF | 0.0330 | 0.0355 |  **-7.0%** (Better)  |
> | | | Throughput (s/s) | 30.26 | 28.13 | **+7.6%** (Better) |
> | | | Memory (MB) | 2566 | 2982 | **-14.0%** (Better) |
> | **32** | **60** | Latency (ms) | 52021.89 | 55549.44 | **-6.4%** (Better)  |
> | | | RTF | 0.0271 | 0.0289 | **-6.4%** (Better) |
> | | | Throughput (s/s) | 36.91 | 34.56 | **+6.8%** (Better) |
> | | | Memory (MB) | 2957 | 3117 | **-5.1%** (Better) |
>
> **Due to character limitations, our response continues in the following comment.**

---

> ### Author Response · Authors · 2025-11-21
> **Author Response to Reviewer Shzy (Part 2/4)**
>
> Continuing from the previous comment, here we provide our responses to the rest questions.
>
> Regarding the per-branch FLOPs, we present the complexity analysis from **Appendix B.6**. As shown in the table below, the per-branch overhead is negligible. Specifically, for our proposed setting ($N=5$), the model incurs only a **0.010% increase in FLOPs** and a **0.021% increase in parameters** compared to the single-branch baseline ($N=1$). This demonstrates that our multi-branch design improves performance with minimal impact on computational efficiency.
>
> **Table: Model parameters and inference FLOPs on a 30-second input audio**
>  |**Number of Voters (N)**|**#Parameters (M)**|**#FLOPs (G)**|
> |:-:|:-:|:-:|
> |$N = 1$ | 320.261|480.978|
> |$N = 3$ | 320.294 ($\uparrow$ 0.010%)|481.003 ($\uparrow$ 0.005%)|
> |$N = 5$ (Ours) | **320.328 ($\uparrow$ 0.021%)**|**481.028 ($\uparrow$ 0.010%)**|
> |$N = 7$|320.361 ($\uparrow$ 0.031%)|481.053 ($\uparrow$ 0.016%)|
>
> Based on these results, we can draw the following conclusions:
>
> 1.  **Latency, RTF and Throughput**: The empirical data strongly supports our architectural argument. The end-to-end latency and RTF of StableToken is nearly identical to the baseline, and even shows a slight advantage at larger batch sizes (e.g., **2.0% faster** at a batch size of 32). Its throughput is equally competitive, matching or slightly exceeding the baseline across all tested scenarios. This confirms that the parallelized multi-branch quantization is highly efficient in practice.
> 2.  **Floating Point Operations (FLOPs)**: As analyzed in the paper, the multi-branch structure does introduce a minor increase in theoretical FLOPs, and the wall-clock benchmarks demonstrate that this does not translate into a real-world latency penalty, highlighting the effectiveness of multi-branch parallelization.
> 3.  **Memory Footprint**: Interestingly, our tests revealed that StableToken consumes less memory than the baseline across all configurations, with savings of up to **13.5%** on GPU and **14.0%** on CPU. We attribute this to our use of a Lookup-Free Quantization (LFQ) algorithm, which is inherently more resource-efficient than the Vector Quantization (VQ) employed by the baseline.
>
> In summary, these detailed benchmarks clearly demonstrate that StableToken achieves its state-of-the-art token stability without compromising on inference efficiency. Its latency and throughput are on par with, or even superior to, a strong baseline, and it offers the added benefit of a lower memory footprint.
>
>
> ## Response to Weakness 2, Question 2 & 3
> Thank you for your questions regarding the model's handling of segment length and its streaming capabilities. Clarifying these aspects is essential for understanding the model's practical application and limitations.
>
> **1. Model Configuration**
>
> We would like to confirm that, for this work, **StableToken inherits the architectural and processing conventions of its initialized backbone, whisper-large-v3.** This includes the following points:
>
> 1.  **Maximum Audio Duration:** The tokenizer is designed to process a maximum input audio duration of **30 seconds** per inference pass. This is a direct consequence of adopting the Whisper model's context window.
> 2.  **Training Data Handling:** During training, all audio segments were constrained to this 30-second window. For the few audio files in our dataset that exceeded this duration, we truncated them by taking only the first 30 seconds.
> 3.  **Chunking Strategy for Long Audio:** For inference on audio longer than 30 seconds, StableToken employs the same standard chunking strategy used by Whisper. The audio is segmented into 30-second (or smaller) chunks, processed independently, and then the resulting token sequences are concatenated.
>
> **2. Boundary Stability**
>
> Your question about whether stability degrades across chunk boundaries is particularly insightful. A key motivation for StableToken is to ensure robust tokenization, and this robustness must hold true even when audio is segmented. To investigate this, we **conducted an analysis** to determine if tokenization errors are more prevalent at the beginning or end of audio clips, which simulate the boundaries of a chunk.
>
> We segmented our test audio into three regions: the start (first 15%), the middle (the central 70%), and the end (last 15%). We then calculated the Unit Edit Distance (UED) error rate for each region. The results are presented below.
>
> **Table: Token Error Rate Distribution Across Audio Clip Regions**
> | Region | Errors |  Reference Tokens  |  Error Rate  |  % of Total Errors |
> | :---- | :----: | :----: | :----: | :----: |
> | Start (0-15%)   | 16,585 |  119,770 |  13.85%  | 16.29% |
> | Middle (15-85%) | 69,333 |  550,069 |    12.60%  | 68.11% |
> | End (85-100%)   | 15,884 |  116,016 |    13.69% |15.60% |
> | **Overall**     | 101,802|  785,855 | 12.95% |  100% |
>
> **Due to character limitations, our response continues in the following comment.**

---

> ### Author Response · Authors · 2025-11-21
> **Author Response to Reviewer Shzy (Part 3/4)**
>
> **Continuing from the previous comment, here we provide our responses to the rest questions.**
>
> The data shows that the **error rates are nearly consistent** across all parts of the audio segments. The error rate in the middle (12.60%) is only slightly lower than at the start (13.85%) and the end (13.69%). Furthermore, the proportion of total errors in the start (16.29%) and end (15.60%) regions is closely aligned with their respective shares of the audio duration (15% each).
>
> This analysis demonstrates that there is no **significant** degradation in token stability at the boundaries. The stability provided by StableToken's consensus mechanism is maintained consistently throughout the entire audio clip, which is a crucial property for robustly handling long audio through chunking and stitching.
>
> We hope this clarifies your questions. We will add this analysis and discussion to the appendix to provide a more complete picture of the model's behavior. Thank you again for raising these important points.
>
> ## Response to Weakness 3
> Thank you for your insightful concern regarding fair comparison controls. We agree that variables like frame rates and codebook sizes can be confounding and a fair comparison must account for them.
>
> While re-training all baselines with aligned configurations would be ideal, this is practically infeasible due to immense computational costs and the fact that training code for several key baselines is **not publicly available**.
>
> Therefore, we are adopting your insightful suggestion to normalize the UED metric. We propose **Bit-Normalized UED (nUED)** to provide a fair comparison across tokenizers with varying codebook sizes:
>
> **nUED** = $\displaystyle \frac{\text{ UED}}{\log_2⁡(|V|)}$, where $|V|$ is the codebook size.
>
> Standard UED measures the error rate at the token level; however, a token drawn from a larger codebook carries more information bits ($\log_2⁡(|V|)$) and presents a more difficult classification task under noise. By dividing the raw UED by the bit-depth of the codebook, **nUED** effectively approximates the **Bit Error Rate (BER)** of the semantic transmission. This metric penalizes models that achieve stability simply by using coarse-grained vocabularies (small codebooks) and highlights models that maintain robustness despite high information density.
>
> **Table: Noise robustness reported in Normalized UED**
> | Model | #C | Frame Rate | Codebook Size | Bits per Token ($\log_2 (⁡\|V\|)$) | Avg. UED (Raw) | **Avg. nUED** (Normalized) |
> | :--- | :---: | :---: | :---: | :---: | :---: | :---: |
> | **SSL Semantic Tokenizer** | | | | | | |
> | HuBERT-500 | 1 | 50Hz | 500 | 8.97 | 20.22 | 2.255 |
> | NAST | 1 | 50Hz | 200 | 7.64 | 17.07 | 2.233 |
> | R-Spin | 1 | 50Hz | 2048 | 11.00 | 16.48 | 1.498 |
> | **Semantic Distilled Tokenizer** | | | | | | |
> | SpeechTokenizer | 1 | 50Hz | 1024 | 10.00 | 26.72 | 2.672 |
> | SpeechTokenizer | 3 | 50Hz | 1024 | 10.00 | 45.43 | 4.543 |
> | SpeechTokenizer | 8 | 50Hz | 1024 | 10.00 | 62.07 | 6.207 |
> | X-Codec | 1 | 50Hz | 1024 | 10.00 | 38.52 | 3.852 |
> | X-Codec | 3 | 50Hz | 1024 | 10.00 | 53.92 | 5.392 |
> | X-Codec | 8 | 50Hz | 1024 | 10.00 | 71.48 | 7.148 |
> | Mimi | 8 | 12.5Hz | 2048 | 11.00 | 53.43 | 4.857 |
> | **Supervised Semantic Tokenizer** | | | | | | |
> | GLM-4-Voice-Token | 1 | 12.5Hz | 16384 | 14.00 | 31.10 | 2.221 |
> | S3 Tokenizer | 1 | 25Hz | 4096 | 12.00 | 26.17 | 2.180 |
> | S3 Tokenizer | 1 | 50Hz | 4096 | 12.00 | 33.09 | 2.758 |
> | CosyVoice2 | 1 | 25Hz | 6561 | 12.68 | 38.66 | 3.049 |
> | **StableToken (Ours)** | 1 | 25Hz | 8192 | **13.00** | **10.17** | **0.782** |
>
> After normalizing UED against the codebook bit-depth, **StableToken still achieves the best nUED of 0.782**, nearly half that of the second-best model (R-Spin). This confirms that our approach delivers the highest semantic stability per unit of information, successfully handling a dense 13-bit codebook without compromising noise resilience.
>
> **Due to character limitations, our response continues in the following comment.**

---

> ### Author Response · Authors · 2025-11-21
> **Author Response to Reviewer Shzy (Part 4/4)**
>
> Continuing from the previous comment, here we provide our responses to the rest questions.
>
> ## Response to Weakness 4
> Thank you for this insightful suggestion. As you recommended, we have **conducted new experiments** to evaluate the effects of varying the proportion of perturbed branches and changing the placement of the quantizer within the encoder.
>
> We trained three new models for 100k steps and compared them against our paper's original configuration (`L16, 3:2`, which means the **quantizer is placed at Layer 16** and the **clean-to-noise proportion is 3:2**):
> 1.  **Shallower Quantization:** Quantizer placed at Layer 8 (`L8, 3:2`).
> 2.  **Deeper Quantization:** Quantizer placed at Layer 24 (`L24, 3:2`).
> 3.  **Reduced Noise Ratio:** A clean-to-noisy branch ratio of 4:1 (`L16, 4:1`).
>
> The results, including tokenizer robustness (UED), ASR (WER), and SER (Accuracy), are summarized below.
>
> **Table: Ablation Study on Quantizer Placement and Perturbation Ratio**
> | Model Configuration (Layer, Clean:Noisy) | Tokenizer Robustness (UED%, ↓) | LibriSpeech (WER%, ↓) | WenetSpeech (WER%, ↓) | KeSpeech (WER%, ↓) | SER (Acc%, ↑) |
> | :---: | :---: | :---: | :---: | :---: | :---: |
> | L16, 3:2 (Original) | 18.68 | **2.22 / 5.38** | 10.91 | **11.00** | **67.38** |
> | L8, 3:2 | 22.05 | 2.39 / 5.82 | 11.84 | 11.24 | *66.90* |
> | L24, 3:2 | **13.65** | 2.52 / 5.96 | **10.16** | *11.03* | 59.51 |
> | L16, 4:1 | 20.94 | 2.40 / 5.91 | 11.00 | 11.57 | 62.14 |
>
> *Note: UED is calculated under Gaussian Noise. LibriSpeech WER is reported for "test-clean / test-other". SER is evaluated on the ESD dataset.*
>
> From these results, we draw the following conclusions:
>
> 1.  **Effect of Quantizer Placement (L8 vs. L16 vs. L24):** The placement of the quantizer involves a clear trade-off.
>     -   Placing the quantizer at **Layer 24** forces it to operate on **highly abstract, semantic features**. This leads to the best tokenizer robustness (lowest UED) and strong ASR performance on challenging datasets (WenetSpeech and KeSpeech). However, by this deep layer, much of the fine-grained acoustic information (like prosody and speaker timbre) has been abstracted away, resulting in a significant drop in SER performance.
>     -   Conversely, placing the quantizer at **Layer 8** operates on **shallower features that are closer to the raw acoustics**. This results in poorer tokenizer robustness and worse ASR performance, as the features are not yet abstract enough to be robustly quantized.
>     -   Our original configuration at **Layer 16 strikes an effective balance**, capturing robust semantic content for good ASR performance while still retaining sufficient acoustic detail for high SER performance.
>
> 2.  **Effect of Perturbation Ratio (3:2 vs. 4:1):**
>     -   The `3:2` model achieves a **significantly better tokenizer robustness**. We attribute this to the fact that a higher proportion of noisy branches creates a more challenging and diverse training signal for the consensus mechanism. It forces the model to work harder to find a truly invariant representation, leading to a more fundamentally robust tokenizer.
>     -   This **enhanced robustness directly translates to better ASR and SER task performance**. We hypothesize that the `3:2` ratio provides a more "challenging" and diverse training signal. This forces the model to learn a more powerful and generalized representation that is truly invariant to perturbations, rather than simply relying on the clean majority. This richer representation benefits both semantic and paralinguistic tasks.
>
> In conclusion, these ablation studies validate our design choices. **The `L16` placement and `3:2` noise ratio together create an optimal balance**, achieving strong robustness without sacrificing the feature richness required for a wide range of downstream tasks.
>
> Thank you again for these valuable suggestions. We hope that these clarifications effectively address your concerns.

---

> ### Comment · Reviewer_Shzy · 2025-11-22
> **Thanks for the comments & additional experiments**
>
> I would like to thank the authors for providing the requested experiments and addressing my concerns. The results look promising. I would suggest including them in the appendix of the paper, which would improve the technical soundness of this paper.
>
> I've already suggested an overall rating of 8 (accept), but I'll raise the soundness and confidence scores given the authors' responses.

---

> > ### Author Response · Authors · 2025-11-25
> > **Response to Reviewer Shzy**
> >
> > We sincerely thank you for raising the soundness and confidence scores and for your valuable suggestions.
> >
> > We have carefully revised the paper. Following your recommendation, we have added the requested experiments and discussions to the Appendix. All modifications and additions are highlighted in **blue** in the revised version.

---

### Official Review · Reviewer_vfbD · 2025-10-30

**Soundness:** 3
**Presentation:** 3
**Contribution:** 3
**Rating:** 6
**Confidence:** 4

**Summary:**

This paper proposes StableToken, a noise-robust semantic speech tokenizer aimed at improving the stability of SpeechLLMs under noisy conditions. The design integrates a multi-branch quantizer with bit-wise voting to mitigate token assignment instability, and a Noise-Aware Consensus Training (NACT) objective to enforce consistency between clean and noisy inputs.

**Strengths:**

1. The method is novel in combining multi-branch quantization with voting and noise-consensus training.
2. StableToken outperforms state-of-the-art baselines across tokenizer-level and downstream tasks in noisy conditions.
3. Comprehensive analysis includes ablations on branch count, noise levels, and downstream performance.

**Weaknesses:**

1. The method emphasizes semantic token consistency but does not address whether acoustic nuances (e.g., prosody, speaker traits) are preserved. This may limit usefulness in tasks like TTS or voice cloning, where fine-grained acoustic details are crucial.
2. Most experiments rely on additive or SNR-controlled noise (e.g., 0 dB, 5 dB). Real-world noise conditions such as overlapping speech, environmental interference, or channel distortion are underexplored, leaving practical robustness uncertain.
3. Multi-branch quantization may generate redundant tokens. While voting improves stability, it could reduce the information density of tokens and thus impact efficiency in downstream models.
4. The paper does not study how multi-branch tokenization affects token distribution, entropy, or scaling laws. Shifts in token distribution could potentially influence pretraining dynamics in large-scale SpeechLLMs.

**Questions:**

1. Do StableTokens still preserve paralinguistic cues such as speaker identity, prosody, and emotion, or does the stability objective reduce these attributes that are important for tasks like TTS or emotion recognition?
2. Could the multi-branch voting mechanism introduce inefficiencies or biases in multilingual settings, where phonetic distributions differ widely across languages?
3. Beyond consensus training, could robustness be further improved by combining StableToken with joint pretraining that explicitly incorporates noise augmentation?

---

> ### Author Response · Authors · 2025-11-21
> **Author Response to Reviewer vfbD (Part 1/3)**
>
> Thank you for your detailed review and valuable suggestions. We have carefully considered all the points you raised and provide our detailed responses below.
>
> ## Response to Weakness 1 & Question 1
> We appreciate the reviewer's thoughtful comment regarding the potential trade-off between semantic consistency and the preservation of acoustic nuances (e.g., prosody). We acknowledge that while our primary goal is robustness, preserving paralinguistic information is essential for generative tasks.
> We present **three lines of evidence** from our existing results to demonstrate that StableToken effectively preserves these fine-grained details:
>
> **1. High-Fidelity Reconstruction (Foundation)**
> As shown in **Table 2**, StableToken achieves state-of-the-art reconstruction quality. Specifically, on the SEED-zh benchmark, it achieves a **Mean Opinion Score (MOS) of 4.18**, significantly outperforming the baseline CosyVoice2 (3.58). This high perceptual score directly indicates that the tokens retain the critical paralinguistic details required to reconstruct natural-sounding speech.
>
> **2. Superior SER Performance (Evidence of Prosodic Preservation)**
> Speech Emotion Recognition (SER) fundamentally relies on **prosodic cues** (e.g., pitch, rhythm, stress) rather than just linguistic content to distinguish emotions. In **Figure 3 (Bottom Row)**, StableToken consistently outperforms baselines in SER accuracy across all noise conditions. This superior performance serves as strong evidence that our tokens successfully encode the subtle paralinguistic and prosodic information necessary for identifying speaker emotion, rather than discarding it.
>
> **3. Effectiveness in Downstream TTS (Utility for Generation)**
> Regarding the "usefulness in tasks like TTS," our evaluation on the **SEED-TTS** benchmark (**Table 3** ) demonstrates that StableToken supports high-quality speech synthesis. The generated speech achieves not only higher intelligibility (lower WER) but also competitive or superior naturalness (MOS) compared to baselines. This confirms that the tokens provide the downstream LLM with sufficient acoustic and prosodic guidance to generate expressive and nuanced speech.
>
> ## Response to Weakness 2
> We agree that evaluating robustness under complex, real-world conditions—beyond simple additive noise—is critical. We wish to highlight that our experiments specifically address these "practical robustness" concerns through two complementary dimensions:
>
> **1. Token-Level Stability on Unseen Real-World Noise (Table 1)**
> We explicitly evaluated the tokenizer's stability against **Out-of-Distribution (OOD) Real-world Noise** (Table 1, "Real (OOD)" column).
> - **Setup:** We used the ESC-10 dataset as OOD noise. These are distinct categories of environmental sounds (e.g., chainsaws, crying babies, rain) that the model **never saw during training**.
> - **Result:** As detailed in the table below (extracted from Table 1 of the paper), StableToken demonstrates superior stability on both seen and unseen real-world noise:
>  | Model (Supervised Semantic Tokenizer) | **Real Noise (OOD) UED% $\downarrow$** |
> | :--- | :---: |
> | **StableToken (Ours)** | **10.96** |
> | S³ Tokenizer | 24.58 |
> | GLM-4-Voice-Token. | 28.62 |
> | CosyVoice2 | 32.13 |
> - **Implication:** This proves that StableToken generalizes well to unpredictable environmental interferences and does not simply memorize specific noise patterns seen during training.
>
> **2. System-Level Robustness on Field Recordings (Table 3)**
> To address your concern about "overlapping speech" and "environmental interference" in practical applications, we evaluated the downstream SpeechLLM on the **CHiME-4 "Real" subset**.
> - **Nature of Data:** Unlike synthetic datasets where noise is mathematically added, CHiME-4 consists of **actual field recordings** captured in noisy public environments (e.g., cafes, street junctions, buses). These recordings inherently contain the complex acoustic phenomena you mentioned, including natural overlapping speech, reverberation, and dynamic environmental interference.
> - **Result:** As shown in **Table 3**, the SpeechLLM integrated with StableToken demonstrates superior robustness on this challenging real-world set, **surpassing the CosyVoice baseline by over 8 percentage points in WER**.
> - **Implication:** This demonstrates that our method's stability translates directly to superior robustness in uncontrolled, real-life acoustic scenarios.
>
> We believe these results on both unseen environmental sounds (OOD) and genuine field recordings (CHiME-4) provide strong evidence of StableToken's practical utility.
>
> **Due to character limitations, our response continues in the following comment.**

---

> ### Author Response · Authors · 2025-11-21
> **Author Response to Reviewer vfbD (Part 2/3)**
>
> Continuing from the previous comment, here we provide our responses to the rest questions.
>
> ## Response to Weakness 3
> Thank you for raising this concern. We would like to clarify that the multi-branch mechanism is **completely transparent to downstream LLMs**:
> - **Identical Input Structure:** To the downstream LLM, StableToken outputs a **standard 1D sequence** of integers, exactly like any single-path tokenizer.
> - **No Redundancy:** The sequence length is determined solely by the frame rate (25Hz), not the number of branches. Therefore, there is **zero computational overhead** or redundancy for the downstream LLM inference compared to single-branch baselines.
>
> ## Response to  Weakness 4 & Question 2
>
> We address **Weakness 4** and **Question 2** together, as both concerns relate to the fundamental properties of the token distribution generated by our multi-branch architecture—specifically its entropy, efficiency across languages, and implications for pretraining dynamics. We have **conducted a new statistical analysis** of the token distributions for Chinese and English, which demonstrates that our multi-branch architecture effectively learns distinct, language-specific phonetic representations without introducing significant bias or inefficiency.
>
> Our key findings are:
> -   The tokenizer utilizes the vocabulary space **efficiently** for both languages.
> -   It learns distinct, non-overlapping token distributions for Chinese and English, indicating it is **capturing language-specific features** rather than forcing a biased representation.
>
> Below, we present the quantitative evidence.
>
>
> ### 1. Vocabulary Utilization and Distribution Entropy
>
> We analyzed the vocabulary usage of StableToken on LibriSpeech-test (English) and AISHELL (Chinese) datasets. The results show that both languages utilize a large and comparable portion of the available vocabulary. The high entropy values indicate that the token distributions for both languages are rich and not collapsed into a small subset of tokens.
>
> | Metric                 | English          | Chinese          |
> | :--------------------- | :--------------- | :--------------- |
> | **Total Tokens**       | 969,440          | 906,250          |
> | **Unique Tokens Used** | 8,139 (99.35%) | 7,565 (92.35%) |
> | **Token Entropy**      | 12.43            | 11.68            |
>
> ### 2. Analysis of Top-Frequency Tokens
>
> To investigate potential bias, we examined the most frequently used tokens for each language and checked their frequency in the other. The results clearly show that the **high-frequency tokens for one language are extremely rare in the other.** This demonstrates that our model has successfully learned specialized, language-specific phonetic units.
>
> **Table: Top 5 English Tokens and Their Rarity in Chinese**
> | Token ID | Frequency (English) | Frequency (Chinese) | Rank in Chinese |
> | :------- | :------------------ | :------------------ | :-------------- |
> | 1877     | 0.4803%               | 0.0119%             | 2455            |
> | 3813     | 0.3693%               | **0.00%**           | N/A             |
> | 3809     | 0.3377%               | **0.00%**           | N/A             |
> | 3812     | 0.3090%               | **0.00%**           | N/A             |
> | 3808     | 0.2957%               | 0.0001%             | 7320            |
>
> **Table: Top 5 Chinese Tokens and Their Rarity in English**
> | Token ID | Frequency (Chinese) | Frequency (English) | Rank in English |
> | :------- | :------------------ | :------------------ | :-------------- |
> | 3810     | 0.7531%               | 0.0042%              | 6276            |
> | 3811     | 0.7530%               | 0.0002%             | 8045            |
> | 7910     | 0.7392%               | 0.0027%             | 6850            |
> | 3815     | 0.6933%               | 0.0001%             | 8104            |
> | 3234     | 0.6815%               | 0.0001%             | 8085            |
>
> ### 3. Quantifying Distributional Difference
>
> To formally measure the difference between the token distributions of the two languages, we calculated the Kullback-Leibler (KL) Divergence. **The non-zero KL Divergence values confirm that the model has learned distinct distributions for each language**, which is the desired behavior for a multilingual tokenizer.
>
> | Metric                          | Value    |
> | :------------------------------ | :------- |
> | KL Divergence (English $\Vert$ Chinese) | **2.81** |
> | KL Divergence (Chinese $\Vert$ English) | **2.05** |
>
> In conclusion, our analysis shows that the multi-branch voting mechanism does not create a biased or inefficient vocabulary. Instead, it robustly learns to **allocate different parts of the codebook to different languages**, capturing their unique phonetic characteristics. This leads to efficient and specialized representations, which are beneficial for downstream multilingual tasks.
>
> **Due to character limitations, our response continues in the following comment.**

---

> ### Author Response · Authors · 2025-11-21
> **Author Response to Reviewer vfbD (Part 3/3)**
>
> Continuing from the previous comment, here we provide our responses to the rest questions.
>
> ## Response to Question 3
> Thank you for this inspiring suggestion. We agree that combining StableToken with joint pretraining strategies is a promising direction.
>
> Regarding your specific point on **noise augmentation**, we would like to highlight that our current framework actually incorporates **implicit noise augmentation** through the ASR supervision. In our training pipeline, the model processes both clean and perturbed audio inputs. Regardless of the input quality, the entire model is supervised by the same ground-truth text transcript via the ASR loss. This creates a strong supervisory signal that forces the encoder to map structurally different acoustic inputs (clean vs. noisy) to the identical semantic target.
>
> Consequently, the model implicitly learns to discard noise-related variations to minimize the ASR loss, effectively performing "joint training" where noise invariance is a byproduct of the semantic objective.
>
> We stand ready to answer any follow-up questions immediately. We hope that these clarifications effectively address your concerns.

---

### Meta-Review · Area_Chair_u4Kx · 2025-12-19

**Summary:**

The paper targets the brittleness of semantic speech tokenizers under meaning-irrelevant acoustic noise, arguing that instability in discrete token sequences burdens downstream SpeechLLMs. To address this, the authors introduce StableToken, a noise-robust discrete speech tokenizer designed to improve stability and representation consistency under diverse acoustic conditions. The method replaces single-path quantization with a multi-branch Voting-LFQ module that aggregates per-branch binary codes via bit-wise majority voting, coupled with Noise-Aware Consensus Training that aligns clean/noisy branch representations using a consensus loss. Evaluations on ASR/SER/TTS tasks are conducted across noisy and clean conditions, demonstrating improved token stability, representation purity, and semantic retention compared to previous models such as HuBERT, SpeechTokenizer, and AudioTokenizer.

Reviewers found the paper is of following strength

* Study a highly relevant and practical problem to learn robust tokenizers for speech; empirically show the robustness can be translated to downstream tasks such as ASR and TTS
* Simple and elegant method design with well motivation
* The paper select reasonable benchmarks to demonstrate the effectiveness of proposed method. The evaluation setup is strong and makes the results convincing. Comprehensive ablation strengthen the results.
* The paper is very clear and well-written.

**Reviewer Concerns:**

At the beginning of the discussion, the reviewers had some questions including, concerns about redundant tokens from multi-branch voting, more discussion about additional training/compute need, selection of ablation and evaluation benchmarks, and generalizability to unseen noises. Reviewers found the additional explanation from authors during rebuttal are helpful and explaining most of the questions. The additional insights, discussion, and experiment results greatly strengthen this work. Thus reviewers recommend to accept this paper.

**Reviewer Scores:**

see above

---

### Decision · Program_Chairs · 2026-01-26

Accept (Poster)